

# Multivariable Integrated Evaluation of Model Performance with the Vector Field Evaluation Diagram

Zhongfeng Xu[1], Ying Han[1], Congbin Fu[2,1]

[1]CAS Key Laboratory of Regional Climate-Environment for Temperate East Asia, Institute of Atmospheric Physics, Chinese
Academy of Sciences, Beijing 100029, China
[2]Institute for Climate and Global Change Research and School of Atmospheric Sciences, Nanjing University, Nanjing, China

*Correspondence to*: Zhongfeng Xu (xuzhf@tea.ac.cn)

**Abstract.** This paper develops a multivariable integrated evaluation (MVIE) method to measure the overall performance of climate model in simulating multiple fields. The general idea of MVIE is to group various scalar fields into a vector field and compare the constructed vector field against the observed one using the vector field evaluation (VFE) diagram. The VFE diagram was devised based on the cosine relationship between three statistical quantities: root mean square length (RMSL) of a vector field, vector field similarity coefficient, and root mean square vector deviation (RMSVD). The three statistical quantities can reasonably represent the corresponding statistics between two multidimensional vector fields. Therefore, one can summarize the three statistics of multiple scalar fields using VFE diagram and facilitate the intercomparison of model performances. The VFE diagram can illustrate how much the overall root mean square deviation of various fields is attributable to the differences in the root mean square value and how much is due to the poor pattern similarity. The MVIE method can be flexibly applied to full fields (including both the mean and anomaly) or anomaly fields depending on the application. We also propose a multivariable integrated evaluation index (MIEI) which takes the amplitude and pattern similarity of multiple scalar fields into account. The MIEI is expected to provide a more accurate evaluation of model performance in simulating multiple fields. The MIEI, VFE diagram, and commonly used statistical metrics for individual variables constitute a hierarchical evaluation methodology, which can provide a more comprehensive evaluation on model performance.

## 1 Introduction

Climate models play a very crucial role in a variety of climate-related studies, e.g., climate dynamics, the detection and attribution of climate change, the projection of future climates and environments, and adaptation to future climate change (IPCC, 2012, 2013). All these studies strongly rely on the performances of climate models. Model evaluation and intercomparison have become increasingly important, especially because a number of climate models are available at present. 29 modelling groups and 60 climate models are involved in the Coupled Model Intercomparison Project Phase 5 (CMIP5) and more are expected to be included in its next phase (Eyring et al., 2016). In addition, more and more regional climate models have been used in regional model downscaling and intercomparison projects (e.g., Fu et al., 2005; Van der Linden



and Mitchell, 2009; Mearns et al., 2009; Giorgi and Gutowski, 2015). Thus, how to concisely summarize and evaluate model performance is extremely important for climate model intercomparison, development, and application.

The Taylor diagram provides a very efficient way to summarize multiple aspects of model performance in simulating scalar
fields (Taylor, 2001). Gleckler et al. (2008) introduced a suite of metrics, e.g., decomposed mean square error, and relative error metrics, which were used to characterize the model performance for various applications. Xu et al. (2016) devised a vector field evaluation (VFE) diagram, which can be regarded as a generalized Taylor diagram, to evaluate the model performance in simulating vector fields, such as vector winds, and temperature gradients. Most metrics, e.g., root mean square error, correlation coefficient, and standard deviation, measure the model performance in simulating an individual
variable (Gleckler et al., 2008). It is a common view that no model performs better than others in every aspect. For example, among various models, one model can show the best performance in simulating air temperature but may have a poor performance in simulating precipitation. In this case, how can researchers select the best model if both temperature and precipitation are important? A popular approach is to show the relative errors of various variables from different models using a portrait diagram (e.g. Gleckler et al. 2008; Pincus, et al. 2008). The portrait diagram illustrates model errors for each
individual variable and can provide an overview of the model performance in simulating various variables. However, the portrait diagram cannot give a quantitative evaluation of the overall performance of climate models in simulating multiple fields. To measure the overall model performance, Gleckler et al. (2008) proposed an exploratory index, termed the model climate performance index (MCPI), by averaging each model's relative errors across multiple fields. Note that the MCPI only considers the root mean square errors (RMSEs) of various fields. The RMSE can be interpreted as a function of the
correlation coefficient and standard deviation (Murphy, 1988; Taylor, 2001; Pincus et al., 2008; Pierce et al., 2009). Therefore, the RMSE takes both the correlation coefficient and standard deviation into account. However, the RMSE cannot explicitly measure the correlation coefficient and standard deviation. For example, the same RMSE can correspond to very different correlation coefficients and standard deviations, especially for large RMSE values.

In this paper, we propose a more comprehensive multivariable integrated evaluation (MVIE) method, which can summarize multiple statistics of model performance in terms of multiple variables, for climate model evaluation. The general idea is to group M scalar fields into an M-dimensional vector field with each dimension representing a scalar field. Such a constructed vector field integrates multiple variables and can be assessed using the VFE diagram. The VFE diagram can concisely summarize the degree of correspondence between simulated and observed vector fields in terms of multiple statistics (Xu et
al, 2016). Therefore, the VFE diagram can be a powerful tool for the MVIE of model performance. To achieve the goal of MVIE, in section 2, we generalize the VFE diagram to evaluate M-dimensional vector fields and interpret three statistical quantities in the VFE diagram from the viewpoint of MVIE. Section 3 presents the approach of MVIE with the VFE diagram. A summary and discussion are provided in section 4.





## 2 Constructing VFE diagram for multidimensional vector fields

Xu et al. (2016) constructed the VFE diagram in terms of 2-dimensional vector fields. There are three statistical quantities in the VFE diagram, i.e., vector similarity coefficient (VSC), root mean square length (RMSL) of a vector field, and root mean square vector deviation (RMSVD) between two vector fields. In this section, each quantity will be defined and interpreted from the viewpoint of MVIE. Thereafter, we will construct the VFE diagram for multidimensional vector fields.

### 2.1 Root mean square length of a vector field

Consider two vector fields $A$ and $B$, which can be spatial fields or temporal fields. Assume that vector fields $A$ and $B$ are derived from a climate model simulation and observation, respectively. Without loss of generality, vector fields $A$ and $B$ can be written as a pair of vector sequences:

$$A_j = (a_{1j}, a_{2j}, ..., a_{Mj}); \qquad j = 1, 2, ..., N$$

$$B_j = (b_{1j}, b_{2j}, ..., b_{Mj}); \qquad j = 1, 2, ..., N$$

Each vector field ($A$ and $B$) consists of $N$ discrete vectors (in time or/and space). Each vector ($A_j$ and $B_j$) has $M$ dimensions. The norms of vectors $A_j$ and $B_j$, the intuitive notion of length, are written as:

$$\|A_j\| = \left( \sum_{i=1}^{M} a_{ij}^2 \right)^{1/2}$$

$$\|B_j\| = \left( \sum_{i=1}^{M} b_{ij}^2 \right)^{1/2}$$

The root mean square lengths (RMSLs) for vector fields $A$ and $B$ are respectively defined as:

$$L_A = \sqrt{\frac{1}{N} \sum_{j=1}^{N} \|A_j\|^2} \qquad (1)$$

and

$$L_B = \sqrt{\frac{1}{N} \sum_{j=1}^{N} \|B_j\|^2} \qquad (2)$$

The square of $L_A$ is rewritten as:



$$L_A{}^2 = \frac{1}{N} \sum_{j=1}^{N} \|\mathbf{A}_j\|^2$$

$$= \frac{1}{N} \sum_{j=1}^{N} \sum_{i=1}^{M} a_{ij}^2$$

$$= \sum_{i=1}^{M} \left( \frac{1}{N} \sum_{j=1}^{N} a_{ij}^2 \right)$$

$$= \sum_{i=1}^{M} L_{ai}^2$$

(3)

where $L_{ai}$ is the root mean square (RMS) value of the $i$-th component of the vector field $\mathbf{A}$. Similarly, we have

$$L_B{}^2 = \frac{1}{N} \sum_{j=1}^{N} \|\mathbf{B}_j\|^2$$

$$= \frac{1}{N} \sum_{j=1}^{N} \sum_{i=1}^{M} b_{ij}^2$$

$$= \sum_{i=1}^{M} \left( \frac{1}{N} \sum_{j=1}^{N} b_{ij}^2 \right)$$

$$= \sum_{i=1}^{M} L_{bi}^2$$

(4)

where

$$L_{ai} = \sqrt{\frac{1}{N} \sum_{j=1}^{N} a_{ij}{}^2}$$

(5)

and

$$L_{bi} = \sqrt{\frac{1}{N} \sum_{j=1}^{N} b_{ij}{}^2}$$

(6)

are the RMS values of the $i$-th component of the vector fields $\mathbf{A}$ and $\mathbf{B}$, respectively. The RMSL of vector field $\mathbf{A}$ reflects the

5   total RMS value across all components of the vector field (Eq. 3). If we break down each variable into its mean and anomaly, it is easy to prove that the mean square value equals the square of the mean plus variance (Eqs. A2, A3). Thus, the RMSL can be used to measure the overall mean value and variance of all components of a vector field. If the vector field is grouped with various scalar fields, the RMSL represents the overall mean value and variance of all scalar fields.



## 2.2 Vector similarity coefficient between two vector fields

In the same as for the vector similarity coefficient (VSC) for 2-dimensional vector fields (Xu et al., 2016), the VSC for M-dimensional vector fields can be defined as:

$$R_v = \frac{\sum_{j=1}^{N} \boldsymbol{A}_j \cdot \boldsymbol{B}_j}{\sqrt{\sum_{j=1}^{N} \|\boldsymbol{A}_j\|^2} \sqrt{\sum_{j=1}^{N} \|\boldsymbol{B}_j\|^2}} \tag{7}$$

The normalized vectors are written as:

5  $\boldsymbol{A}_j^* = \frac{\boldsymbol{A}_j}{L_A} = (a_{1j}^*, a_{2j}^*, \ldots, a_{Mj}^*) ; \qquad j = 1, 2, \ldots, N$

$\boldsymbol{B}_j^* = \frac{\boldsymbol{B}_j}{L_B} = (b_{1j}^*, b_{2j}^*, \ldots, b_{Mj}^*) ; \qquad j = 1, 2, \ldots, N$

With the aid of Eqs. (1) and (2), we have

$$\sum_{j=1}^{N} \|\boldsymbol{A}_j^*\|^2 = \sum_{j=1}^{N} \|\boldsymbol{B}_j^*\|^2 = N \tag{8}$$

We can also represent Eq. (7) in the following form:

$$\begin{aligned} R_v &= \frac{1}{N} \sum_{j=1}^{N} \boldsymbol{A}_j^* \cdot \boldsymbol{B}_j^* \\ &= \frac{1}{N} \sum_{j=1}^{N} \sum_{i=1}^{M} a_{ij}^* b_{ij}^* \end{aligned} \tag{9}$$

VSC can be interpreted as the mean of inner products between paired- and normalized-vectors $\boldsymbol{A}_j^*$ and $\boldsymbol{B}_j^*$. The squared

10  Euclidean Distance (SED) between $\boldsymbol{A}_j^*$ and $\boldsymbol{B}_j^*$ is defined as follows:

$$\|\boldsymbol{C}_j^*\|^2 = \|\boldsymbol{A}_j^* - \boldsymbol{B}_j^*\|^2 \tag{10}$$

With the aid of Eqs. (9) and (10), the sum of all SEDs can be written as:

$$\begin{aligned} \sum_{j=1}^{N} \|\boldsymbol{C}_j^*\|^2 &= \sum_{j=1}^{N} \|\boldsymbol{A}_j^* - \boldsymbol{B}_j^*\|^2 \\ &= \sum_{j=1}^{N} \sum_{i=1}^{M} (a_{ij}^* - b_{ij}^*)^2 \\ &= \sum_{j=1}^{N} \left( \sum_{i=1}^{M} a_{ij}^{*2} + \sum_{i=1}^{M} b_{ij}^{*2} - 2 \sum_{i=1}^{M} a_{ij}^* b_{ij}^* \right) \\ &= \sum_{j=1}^{N} \|\boldsymbol{A}_j^*\|^2 + \sum_{j=1}^{N} \|\boldsymbol{B}_j^*\|^2 - 2 N \cdot R_v \end{aligned}$$

With the aid of Eq. (8), we obtain





$$R_v = 1 - \frac{1}{2N} \sum_{j=1}^{N} \left\| C_j^* \right\|^2 \tag{11}$$

Given the triangle inequality, $0 \le \left\| C_j^* \right\| \le \left\| A_j^* \right\| + \left\| B_j^* \right\|$, we have

$$0 \le \left\| C_j^* \right\|^2 \le \left( \left\| A_j^* \right\| + \left\| B_j^* \right\| \right)^2 \le 2 \left\| A_j^* \right\|^2 + 2 \left\| B_j^* \right\|^2.$$

Adding all SEDs together yields

$$0 \le \sum_{j=1}^{N} \left\| C_j^* \right\|^2 \le 2 \sum_{j=1}^{N} \left\| A_j^* \right\|^2 + 2 \sum_{j=1}^{N} \left\| B_j^* \right\|^2 = 4N \tag{12}$$

Substituting Eq. (12) into Eq. (11), we obtain $-1 \le R_v \le 1$. Thus, the VSC between two M-dimensional vector fields varies

5    from -1 to 1. The VSC reaches it maximum of 1 when each pair of normalized vectors has exactly the same length and direction, i.e., $A_j^* = B_j^*$ for all $i$ ($1 \le i \le N$). The VSC reaches it minimum value of -1 when each pair of normalized vectors has exactly the same length but points in opposite direction, i.e., $A_j^* = -B_j^*$ for all $i$ ($1 \le i \le N$).

With the aid of Eqs. (1) and (2), Eq. (7) can be written as:

$$
\begin{aligned}
R_v &= \frac{1}{N L_A L_B} \sum_{j=1}^{N} A_j \cdot B_j \\
&= \frac{1}{N L_A L_B} \sum_{j=1}^{N} \sum_{i=1}^{M} a_{ij} b_{ij} \\
&= \frac{1}{N L_A L_B} \sum_{j=1}^{N} \sum_{i=1}^{M} \frac{a_{ij} b_{ij}}{\sigma_{ai} \sigma_{bi}} \sigma_{ai} \sigma_{bi} \\
&= \frac{1}{N L_A L_B} \sum_{i=1}^{M} \sum_{j=1}^{N} \frac{a_{ij} b_{ij}}{\sigma_{ai} \sigma_{bi}} \sigma_{ai} \sigma_{bi} \\
&= \frac{1}{L_A L_B} \sum_{i=1}^{M} \left( \sigma_{ai} \sigma_{bi} \frac{1}{N} \sum_{j=1}^{N} \frac{a_{ij} b_{ij}}{\sigma_{ai} \sigma_{bi}} \right) \\
&= \frac{1}{L_A L_B} \sum_{i=1}^{M} \sigma_{ai} \sigma_{bi} R_{ui}
\end{aligned}
\tag{13}
$$

10    where $\sigma_{ai} = \sqrt{\frac{1}{N} \sum_{j=1}^{N} a_{ij}^2}$ and $\sigma_{bi} = \sqrt{\frac{1}{N} \sum_{j=1}^{N} b_{ij}^2}$ are the uncentered RMS values of the $i$-th component of vector fields $A$ and

$B$. $R_{ui} = \frac{\frac{1}{N} \sum_{j=1}^{N} a_{ij} b_{ij}}{\sigma_{ai} \sigma_{bi}}$ is the uncentered pattern correlation coefficient between the $i$-th paired components of vector fields $A$ and $B$. The uncentered pattern correlation coefficient is a variant of Pearson's correlation in which the mean values are not





removed. $R_{ui}$ can also be interpreted as the normalized inner product of two $N$-dimensional vectors $\boldsymbol{a}_i = (a_{i1}, a_{i2}, \dots, a_{iN,})$ and $\boldsymbol{b}_i = (a_{i1}, a_{i2}, \dots, a_{iN,})$:

$$R_{ui} = \frac{<\boldsymbol{a}_i \cdot \boldsymbol{b}_i>}{\|\boldsymbol{a}_i\|\|\boldsymbol{b}_i\|} = \frac{\sum_{j=1}^{N} a_{ij} b_{ij}}{\sqrt{\sum_{j=1}^{N} a_{ij}^2}\sqrt{\sum_{i=1}^{N} b_{ij}^2}} \tag{14}$$

The uncentered correlation coefficient can be interpreted as the cosine of the angle between the $N$-dimensional vectors $\boldsymbol{a}_i$ and $\boldsymbol{b}_i$. $R_{ui}$ increases when the arguments of vectors $\boldsymbol{a}_i$ and $\boldsymbol{b}_i$ approach each other (Eq. 14). Thus, the similarity coefficient

5 between two vector fields $A$ and $B$ can be interpreted as a weighted average of uncentered correlation coefficients across all paired components between two vector fields (Eq. 13).

**2.3 Root mean square vector deviation**

To measure the difference in vector fields $A$ and $B$, a root mean square vector deviation (RMSVD) is defined as:

$$RMSVD = \left[\frac{1}{N}\sum_{j=1}^{N}\|A_j - B_j\|^2\right]^{\frac{1}{2}}$$

$$= \left[\frac{1}{N}\sum_{j=1}^{N}\sum_{i=1}^{M}(a_{ij} - b_{ij})^2\right]^{\frac{1}{2}} \tag{15}$$

10 The square of the RMSVD can be written as:

$$RMSVD^2 = \frac{1}{N}\sum_{j=1}^{N}\sum_{i=1}^{M}(a_{ij} - b_{ij})^2$$

$$= \sum_{i=1}^{M}\left(\frac{1}{N}\sum_{j=1}^{N}(a_{ij} - b_{ij})^2\right) \tag{16}$$

$$= \sum_{i=1}^{M} RMSD_i^2$$

where $RMSD_i = \frac{1}{N}\sum_{j=1}^{N}(a_{ij} - b_{ij})^2$ is the root mean square deviation (RMSD) between the $i$-th paired component of vector fields $A$ and $B$. Thus, the RMSVD measures the overall RMSDs of all components between the original vector fields $A$ and $B$.



## 2.4 Construction of VFE diagram for M-dimensional vector fields

With the aid of Eq. (7), the square of the RMSVD can be written as:

$$
\begin{aligned}
RMSVD^2 &= \frac{1}{N}\sum_{j=1}^{N}\left\|\boldsymbol{A}_j - \boldsymbol{B}_j\right\|^2 \\
&= \frac{1}{N}\sum_{j=1}^{N}\left(\left\|\boldsymbol{A}_j\right\|^2 + \left\|\boldsymbol{B}_j\right\|^2 - 2\boldsymbol{A}_j\cdot\boldsymbol{B}_j\right) \\
&= \frac{1}{N}\sum_{j=1}^{N}\left\|\boldsymbol{A}_j\right\|^2 + \frac{1}{N}\sum_{j=1}^{N}\left\|\boldsymbol{B}_j\right\|^2 - 2R_v\cdot\sqrt{\frac{1}{N}\sum_{j=1}^{N}\left\|\boldsymbol{A}_j\right\|^2}\sqrt{\frac{1}{N}\sum_{j=1}^{N}\left\|\boldsymbol{B}_j\right\|^2}
\end{aligned}
\tag{17}
$$

With the aid of Eqs. (1), (2), and (7), Eq. (17) can be written as:

$$
RMSVD^2 = L_A^2 + L_B^2 - 2R_v\cdot L_A L_B
\tag{18}
$$

The RMSVD, $L_A$, $L_B$, and $R_v$ are related by the law of cosines (Eq. 18). We can construct the VFE diagram for M-dimensional vector fields based on Eq. (18). The VFE diagram and the geometric relationship between $L_A$, $L_B$, $R_v$, and the RMSVD are shown in Fig. 1. As for the case of 2-dimensional vectors (Xu et al., 2016), the RMSLs, i.e., $L_A$ and $L_B$, measure the mean and variance of the lengths of vector fields $A$ and $B$, respectively (Eqs. A2, A3). $R_v$ reflects the pattern similarity between two vector fields. The RMSVD describes the overall difference between two vector fields. Thus, three statistical
quantities can be indicated by a single point on the VFE diagram (Fig. 1).

## 3 Multivariable integrated evaluation with the VFE diagram

### 3.1 Methodology

To evaluate model performance in terms of the simulation of multivariables, one can group various scalar fields into a vector field and compare the constructed vector field against the observed one using the VFE diagram. For example, we can
construct a vector field with temperature and precipitation as its x- and y-component, respectively. One can certainly use more variables as needed to construct the vector field. Given the differences in units and order of magnitude of various variables, we need to normalize all variables before grouping them into a vector field. The normalization can be done by dividing the RMS value of each observational estimate as follows:

$$
\boldsymbol{A}_j^\star = \left(\frac{a_{1j}}{L_{b1}}, \frac{a_{2j}}{L_{b2}}, \ldots, \frac{a_{Mj}}{L_{bM}}\right) = (a_{1j}^\star, a_{2j}^\star, \ldots, a_{mj}^\star); \qquad j = 1, 2, \ldots, N
\tag{19}
$$

$$
\boldsymbol{B}_j^\star = \left(\frac{b_{1j}}{L_{b1}}, \frac{b_{2j}}{L_{b2}}, \ldots, \frac{b_{Mj}}{L_{bM}}\right) = (b_{1j}^\star, b_{2j}^\star, \ldots, b_{Mj}^\star); \qquad j = 1, 2, \ldots, N
\tag{20}
$$



where $L_{bi} = \sqrt{\frac{1}{N}\sum_{j=1}^{N} b_{ij}^{2}}$ is the RMS value for the *i*-th component of vector field $\boldsymbol{B}$ obtained from observational estimates.

Each component of the normalized vector field is dimensionless and on the order of 1. Thus, the statistics of each component are equally important to the total statistics of the vector fields. The normalization is especially necessary when the variables are of different orders of magnitude. For example, the surface air temperature (SAT) is typically on the order of $10^{2}$ K, but
precipitation is generally on the order of $10^{-5}$–$10^{-4}$ mm s$^{-1}$. Under this circumstance, the differences in the VSC, RMSL, and RMSVD between various models would be primarily determined based on the SAT and barely impacted by the precipitation if no normalization was applied. Therefore, in terms of the MVIE of the model performance, the VSC, RMSLs, and RMSVD should be computed using the normalized vector fields $\boldsymbol{A}^{\star}$ and $\boldsymbol{B}^{\star}$. As interpreted in section 2, three statistical quantities in the VFE diagram represent the overall statistics across all components between two vector fields. If the vector fields are
grouped by various scalar fields, the VFE diagram can summarize the three statistics of model performance in simulating multiple scalar fields.

**3.2 Application of multivariable integrated evaluation of model performance**

Without loss of generality, we choose the climatological mean SAT and precipitation as well as the standard deviation of the SAT and precipitation as the variables to interpret the MVIE method. Four variables derived from climate models are
examined against the corresponding observational estimates. The evaluation is based on the monthly mean datasets derived from the first ensemble run of CMIP5 historical experiments during the period from 1961 to 2000 (Taylor, 2012). Three pairs of observed SAT and precipitation datasets are used in this study. The first pair of dataset is the Climatic Research Unit (CRU) gridded SAT and precipitation (Harris, et al., 2014). The second pair of dataset is the University of Delaware air temperature and precipitation (Willmott and Matsuura, 2001). The third pair of dataset is composed of the Global Historical
Climatology Network (GHCN) temperature (Fan and van den Dool, 2008) and Global Precipitation Climatology Centre (GPCC) precipitation (Schneider et al., 2014). All observational data are available at 0.5°×0.5° resolution. We take the average of three pairs of SAT and precipitation values as the reference data in this study. The observational uncertainty can be roughly estimated by comparing each observational estimate to the reference data (Xu et al., 2016). All datasets were regridded to a common resolution of 2.5°×2.5° using a box averaging (bi-linear interpolation) method that re-grids data to a
coarse (finer) resolution. Both the model and observational data are normalized by the RMS value of each observed field before computing their statistics (Eqs. 17, 18).

Table 1 shows the various statistics of 9 CMIP5 models in terms of the climatological mean SAT, precipitation, and the temporal standard deviation of SAT and precipitation over the global land area (60°S–60°N). The standard deviation reflects
the amplitude of interannual variation. The models can generally well simulate the climatological mean SAT characterized by the close correspondence of the RMS values, high uncentered correlation, and small RMSD between the model and observation. In contrast, models show a relatively poor performance in simulating other variables, i.e., climatological mean



precipitation, standard deviations of SAT and precipitation. These statistics vary from one model to the next. It is difficult to compare the overall performances of various models because there are too many variables and models to distinguish one from another (Table 1). It is very useful to summarize the statistics of multiple variables with fewer indices, which enables an objective evaluation of the overall model performance in simulating multiple variables. To achieve this goal, we first

normalized the modeled and observed climatology for summer SAT and precipitation and the temporal standard deviations of the SAT and precipitation (Eqs. 17, 18). Second, the four normalized scalar fields are grouped into a four-dimensional vector field. Third, we computed the statistical quantities RMSL, VSC, and RMSVD statistical quantities with the four-dimensional vector fields derived from model and observational data. As interpreted in section 2, the RMSL (RMSVD) measures the overall RMS values (RMSDs) of all scalar fields (Eqs. 3, 16). The VSC represents the weighted average of

uncentered correlation coefficients across all scalar fields (Eq. 13). Thus, the model performance in simulating multiple variables can be summarized by three statistical quantities. Lastly, the three quantities, i.e., RMSL, VSC, and RMSVD, can be represented by a single point for each model on the VFE diagram. The single point is determined by 12 statistical quantities (4 variables × 3 statistics) those derived from various scalar fields (Table 1). Thus, each point on the VFE diagram can represent the overall performance of an individual model in terms of 3 statistical quantities derived from multiple scalar

fields (Fig. 2).

As shown in Fig. 2, the VSC varies from 0.90 to 0.95, indicating which models can better reproduce the overall spatial pattern of various variables and which cannot. For example, model 1 shows the maximum VSC, indicating that model 1 can generally better reproduce the spatial pattern of the four variables relative to other models. This can be confirmed by Table 1.

The uncentered pattern correlation coefficients for the four scalar fields are generally higher in model 1 than in the other models. Fig. 2 also clearly shows which model overestimates or underestimates the overall RMS values. For example, models 5 and 7 overestimate the RMSLs of the four-dimensional vector fields, suggesting that both models generally overestimate the RMS values of the four scalar fields. This can also be confirmed by Table 1, as model 5 clearly overestimates the RMS values of Ta (1.42) and Pa (1.16) and slightly underestimates the RMS values of Tm (0.99) and Pm

(0.94). Model 7 overestimates all RMS values (1.08, 1.10, 1.11, and 1.07) of the four variables. Thus, the RMSL of a constructed vector field can reasonably represent the overall performance of a model in reproducing RMS values of multiple scalar fields. In contrast, model 9 clearly underestimates the RMSL of the vector field (Fig. 2). Correspondingly, three out of the four RMS values of scalar fields are smaller than 1 for model 9 (Table 1). Similarly, the RMSVD between two vector fields can also reasonably represent the overall RMSDs of multiple scalar fields as shown in Fig. 2 and Table 1. Thus, one

can evaluate the model performance in simulating multiple variables with three statistical quantities. The three statistical quantities represent different aspects of model performance, the knowledge of which can provide a more comprehensive model evaluation. The VFE diagram can clearly illustrate to what extent the overall RMSDs of various scalar fields (represented by the RMSVD) are attributable to the systematic difference in RMS values (represented by the RMSL) and how much is due to the poor pattern similarities (represented by $R_v$).





Model performance changes monotonically with the increase in the correlation coefficient (decrease in the RMSD) if the other statistical quantities remain unchanged. Unlike the correlation coefficient and RMSD, model performance does not change monotonically with the increase or decrease in RMS values. Specifically, model performance improves as the

normalized RMS values approach 1 but decreases as the normalized RMS values approach either zero or infinity. As defined in Eq. (3), the RMSL is equal to the sum of RMS values of all components of a vector field. Thus, RMSL=$N$, i.e., the modeled RMSL is equal to the observed one, does not necessarily suggest that the model well reproduces the RMS values of various scalar fields. This conclusion may also result from the cancellation between the overestimated and the underestimated RMS values. For example, as shown in Table 1, model 3 overestimates the RMS values of Tm (1.06) and Ta

(1.27) but underestimates the RMS values of Pm (0.81) and Pa (0.79). However, the RMSL is consistent with the observational estimate (1.00). Under such a circumstance, the RMSL misrepresents the model performance in simulating RMS values of various scalar fields. To mitigate this shortcoming, one can add a line segment centered at each plotted point along the azimuthal direction (Fig. 3). The length of the line segment is equal to the standard deviation of RMS values of multiple scalar fields. Thus, the length of the line segment can measure the dispersion of various RMS values relative to their

mean. A shorter line indicates that the RMS values are close to the mean. In contrast, a longer line segment indicates that the RMS values are spread out over a wider range. To measure the accuracy of modeled RMS values to that of those observed, one can use the root mean square deviation of the RMS values of various variables:

$$RMSD_L^2 = \frac{1}{M} \sum_{i=1}^{M} (L_{ai}^\star - L_{bi}^\star)^2 \tag{21}$$

where $L_{ai}^\star = \frac{1}{L_{bi}} \sqrt{\frac{1}{N} \sum_{j=1}^{N} a_{ij}^2}$ and $L_{bi}^\star = \frac{1}{L_{bi}} \sqrt{\frac{1}{N} \sum_{j=1}^{N} b_{ij}^2}$ are the RMS values of the $i$-th normalized component of vector fields $\boldsymbol{A}$ and $\boldsymbol{B}$, respectively. With the support of Eq. (6), we have $L_{bi}^\star = 1$ for all $i$ ($1 \leq i \leq$ M). The $RMSD_L^2$ can be further

written as:

$$
\begin{aligned}
RMSD_L^2 &= \frac{1}{M} \sum_{i=1}^{M} (L_{ai}^\star - 1)^2 \\
&= \frac{1}{M} \sum_{i=1}^{M} L_{ai}^{\star\,2} - \frac{2}{M} \sum_{i=1}^{M} L_{ai}^\star + 1 \\
&= \frac{1}{M} \sum_{i=1}^{M} \left( \overline{L_a^\star} + L_{ai}^{\star\,\prime} \right)^2 - 2\overline{L_a^\star} + 1 \\
&= \overline{L_a^\star}^2 + \frac{1}{M} \sum_{i=1}^{M} L_{ai}^{\star\,\prime\,2} - 2\overline{L_a^\star} + 1 \\
&= (\overline{L_a^\star} - 1)^2 + \sigma_{RMS}^2
\end{aligned}
\tag{22}
$$





where $\overline{L_a^\star}$, and ${L_{ai}^\star}'$ are the mean and anomaly of $L_{ai}^\star$, respectively. The mean square value of ${L_{ai}^\star}'$ is written as follows:

$$\sigma_{RMS}^2 = \frac{1}{M}\sum_{i=1}^{M}{L_{ai}^\star}'^2 \tag{23}$$

$\sigma_{RMS}^2$ can also be interpreted as the centered mean square value or the variance of $L_{ai}^\star$. Thus, the $RMSD_L$ can be interpreted as the mean error plus the variance of RMS values of normalized scalar fields (Eq. 22). $RMSD_L$ measures the overall deviation

of modeled RMS values from the observed ones. The modeled RMS values of various scalar fields are exactly equal to the corresponding observed ones only when the $RMSD_L$ is equal to 0.

### 3.3 Multivariable integrated evaluation index for model performance

In general, the model results get closer to the observational estimate as the RMSVD decreases. It is noteworthy that for a given VSC at a relatively low value, the RMSVD does not strictly decrease monotonically as the simulated RMSL

approaches the observed one (Fig. 4). For example, model B shows the same VSC as that of Model A but a smaller bias in the RMSL, which suggest that model B performs better than model A. However, the RMSVD is greater in model B than in model A (Fig. 4). Thus, the decrease in the RMSVD may not necessarily indicate an improvement in model performance. On the other hand, given the drawback of the RMSL in measuring the accuracy of RMS values, the model skill score, defined based on the RMSL and VSC in Xu et al. (2016), is also not well suited for measuring the model performance in simulating

multiple scalar fields. To better measure model performance, we define a multivariable integrated evaluation index (MIEI) based on the VFE diagram (Fig. 4):

$$MIEI^2 = BC^2 + BG^2$$

Based on the law of cosines, we have

$$BG^2 = 2 - 2R_v$$

Thus, the MIEI can be written as:

$$\begin{aligned} MIEI^2 &= RMSD_L^2 + 2(1 - R_v) \\ &= \sigma_{RMS}^2 + (\overline{L_a^\star} - 1)^2 + 2(1 - R_v) \end{aligned} \tag{24}$$

Clearly, the MIEI takes both the amplitudes and pattern similarities of various variables into account and therefore can

provide a comprehensive evaluation of model performance (Eq. 24). In comparison with the RMSVD, the MIEI satisfies the monotonic property of an index with respect to model performance. Specifically, for any given $\sigma_{RMS}$ and $\overline{L_a^\star}$, the MIEI decreases monotonically with the increase in $R_v$. For any given $\sigma_{RMS}$ and $R_v$, the MIEI decreases monotonically as $\overline{L_a^\star}$ approaches 1. For any given $\overline{L_a^\star}$ and $R_v$, the MIEI decreases monotonically with the decrease in $\sigma_{RMS}$. The MIEI is equal to 0 only when $\sigma_{RMS}=0$, $\overline{L_a^\star}=1$, and $R_v=1$, which define a perfect model. In other words, modeled multiple fields are exactly the

same as the observed ones when the MIEI is equal to 0.



As interpreted in section 2, the RMSVD is determined based on the sum of quadratic RMSDs of various scalar fields (Eq. 16). Thus, the RMSVD is equivalent to the model climate performance Index used in previous studies (e.g., Gleckler et al., 2008; Radić and Clarke, 2011; Chen and Sun, 2015). In general, both the RMSVD and MIEI can be used to measure the model performance. However, the MIEI is expected to provide a more accurate evaluation of model performance than the

RMSVD. For example, model 3 shows a smaller RMSVD but a larger MIEI relative to model 2 (Table 1, Fig. 3). The RMSVD and MIEI give an opposite rank in the performances of models 2 and 3. Note that model 3 shows a much greater standard deviation of RMS values (0.23) than that of model 2 (0.07), suggesting that model 3 poorly simulates the relative amplitude of the four variables. Such information is not considered by the RMSVD but can be captured by the MIEI (Eq. 21). The values of the MIEI derived from various models are also shown in Fig. 3. A smaller MIEI generally indicates a better

performance of the climate model. For example, models 1 and 6 show smaller MIEIs than those of other models. Models 1 and 6 show higher VSC values, smaller $RMSD_L$ values, and a close correspondence between RMS values relative to the observed ones (Fig. 3). The MIEI can serve as an index to determine the rank of climate model performance in simulating multiple fields. In comparison with the MIEI, the VFE diagram can provide more comprehensive statistics on the model performance, i.e., pattern similarity, RMS values and their variances, and RMSVD.

**4 Summary and discussion**

The multivariable integrated evaluation (MVIE) method proposed here provides a way of representing the statistics of multiple fields with a single point on a two-dimensional plot, i.e., the VFE diagram. The VFE diagram includes three statistical quantities, i.e., RMSL, VSC, and RMSVD, representing different aspects of model performance, and therefore can provide a more comprehensive evaluation. Specifically, the RMSL (RMSVD) represents the total mean value and variance

(total RMSDs) of all scalar fields. The VSC measures the overall pattern similarity across all scalar fields. As shown in the example, each of the three statistical quantities can reasonably represent the corresponding statistics of multiple scalar fields. Therefore, the VFE diagram can illustrate how much the overall RMSD of various fields is attributable to the difference in RMS values and how much is due to poor pattern similarity. Thus, one can summarize multiple statistics of multivariables for various models in a diagram and facilitate the intercomparison of model performances in simulating multiple variables.

The MVIE method can be applied to spatial or temporal fields. It can also simultaneously evaluate various temporal variabilities simulated by models, e.g., climatological mean state and the amplitude of interannual variability as shown in section 3.2. Based on the VFE diagram, we also developed a multivariable integrated evaluation index (MIEI) which takes the amplitude and pattern similarity of multiple fields into account. The MIEI satisfies the criterion that a model performance index should vary monotonically as the model performance improves. The MIEI provides a more concise evaluation than the

VFE diagram of model performance in simulating multiple fields.



The statistical metrics presented in this paper can be divided into three different levels and their relationships are summarized in a pyramid chart (Fig. 5). The first level of metrics, i.e., correlation coefficient, RMS value, and RMSD, measures model performance in terms of individual variables. These metrics can be illustrated by a table of metrics (Table 1) or a portrait diagram, which can provide detailed information on model performance in simulating individual variables but

cannot give a quantitative evaluation of the overall model performance in simulating multiple fields. The second level of metrics, i.e., the VSC, RMSL, variance of RMSL, and RMSVD, are derived from the first level of metrics and represent the overall statistics of multiple variables. The second level of metrics can be presented as a VFE diagram, which provides an integrated evaluation of model performance in terms of simulating multiple fields. The MIEI belongs to the third level of metrics, which is defined based on the VFE diagram. The MIEI further summarizes the three statistical quantities of the VFE

diagram into a single index and can be used to rank the performances of various climate models. A higher level of metrics provides a more concise evaluation of model performance compared to a lower level of metrics, which facilitates model intercomparison. Unavoidably, a higher level of metrics loses detailed statistical information in contrast to a lower level of metrics. To provide a more comprehensive evaluation of model performance, one can show the VFE diagram together with a table of statistical metrics (Table 1) or other model performance metrics as needed.

As shown in section 2, the VFE diagram can be constructed by using uncentered statistics, which are computed using the full scalar fields, including both mean and anomaly. The VFE diagram can also be computed by uing centered statistics (Appendix A). The centered RMSL of a vector represents the overall variance of all components of a vector field (Eq. A2). The centered VSC can be interpreted as weighted Pearson's correlation coefficients, which measures the overall pattern

similarity across all paired anomaly fields (Eq. A7). The centered RMSVD is equal to the sum of centered RMSDs across all paired components between two vector fields. The type of statistics, i.e., centered or uncentered statistics, that should be used depends on the application. The uncentered statistics should be used if both the mean and anomaly need to be evaluated. In contrast, the centered statistics should be used if the anomaly fields are the primary concern. The centered correlations alone are not sufficient for detection studies (Legates and Davis, 1997). It has been argued that the uncentered statistics are better

suited for detection because they incorporate the response of the mean value. In contrast, the centered statistics are more appropriate for attribution because they better measure the similarity between spatial patterns (Hegerl et al., 2001). The VFE diagram provides us flexibility in model evaluation. In terms of model evaluation aimed at a detection study, one can compute three uncentered statistics with full fields. In contrast, one can use centered statistics by computing three statistical quantities with anomaly vector fields if an attribution study is the major concern of model evaluation.

In practice, one may want to weight different fields based on their relative importance. Determining the weight coefficient depends on the application and therefore is beyond the scope of this study. Here, we only discuss how the weight can be considered in the multivariable integrated evaluation. The MVIE method presented in this study requires the normalization of each modeled and observed variable by dividing the corresponding RMS value of the observed variable (Eqs. 19, 20).



Therefore, one should weight different variables after the normalization (Eqs. B1, B2); otherwise the normalization process will remove the weight coefficient. Weighting each normalized field leads to a quadratic weighting of the quadratic RMS values, quadratic RMSDs, and correlation coefficient (Eqs. B1, B6, B10, B13).

5     The VFE diagram and MIEI may also provide some guidance in weighting various climate models to constrain future climate projection. A recent study suggested that model weighting should take both model performances and model interdependencies into account to improve climate projections (Knutti et al., 2017). The VFE diagram proposed in this paper can summarize model performances in terms of multiple statistics of multivariables on one hand. On the other hand, the VFE diagram can also clearly show the differences between model and observation as well as the differences between various

10     models. These information provided by the VFE diagram may be used in weighting climate models, which warrant for further studies.

**Code availability**

The code used in the production of Figure 3 and Table 1 are available in the supplement to the article.



## Appendix A: Decomposition of RMSL, VSC, and RMSVD

To further interpret the RMSL, VSC, and RMSVD, we break down the full vector fields $A$ and $B$ into the mean and anomaly:

$$A_j = \overline{A} + A'_j = \left(\overline{a}_1 + a'_{1j},\ \overline{a}_2 + a'_{2j}, \cdots, \overline{a}_m + a'_{mj}\right); \qquad j = 1, 2, \ldots, N$$

$$B_j = \overline{B} + B'_j = \left(\overline{b}_1 + b'_{1j},\ \overline{b}_2 + b'_{2j}, \cdots, \overline{b}_m + b'_{mj}\right); \qquad j = 1, 2, \ldots, N$$

where

$$\overline{A} = (\overline{a}_1, \overline{a}_2, \cdots, \overline{a}_m),$$

$$\overline{B} = \left(\overline{b}_1, \overline{b}_2, \cdots, \overline{b}_m\right),$$

$$A'_j = \left(a'_{1j}, a'_{2j}, \cdots, a'_{mj}\right),$$

$$B'_j = \left(b'_{1j}, b'_{2j}, \cdots, b'_{mj}\right),$$

$$\overline{a}_i = \frac{1}{N}\sum_{j=1}^{N} a_{ij}; \qquad i = 1, 2, \ldots, M$$

$$\overline{b}_i = \frac{1}{N}\sum_{j=1}^{N} b_{ij}; \qquad i = 1, 2, \ldots, M$$

$$a'_{ij} = a_{ij} - \overline{a}_i; \qquad i = 1, 2, \ldots, M$$

$$b'_{ij} = b_{ij} - \overline{b}_i; \qquad i = 1, 2, \ldots, M$$

The RMSL of vector field $A$ is written as follows:

$$L_A{}^2 = \frac{1}{N}\sum_{j=1}^{N}\|A_j\|^2$$

$$= \frac{1}{N}\sum_{j=1}^{N}\sum_{i=1}^{M}\left(\overline{a}_i + a'_{ij}\right)^2$$

$$= \frac{1}{N}\sum_{j=1}^{N}\sum_{i=1}^{M}\overline{a}_i{}^2 + \frac{1}{N}\sum_{j=1}^{N}\sum_{i=1}^{M}a'_{ij}{}^2 + \frac{1}{N}\sum_{j=1}^{N}\sum_{i=1}^{M}2\overline{a}_i a'_{ij}$$

Given $\sum_{j=1}^{N} a'_{ij} = 0$, $L_A{}^2$ can be written as:

$$L_A{}^2 = \frac{1}{N}\sum_{j=1}^{N}\sum_{i=1}^{M}\overline{a}_i{}^2 + \frac{1}{N}\sum_{j=1}^{N}\sum_{i=1}^{M}a'_{ij}{}^2$$

$$= \frac{1}{N}\sum_{j=1}^{N}\|\overline{A}\|^2 + \frac{1}{N}\sum_{j=1}^{N}\|A'_j\|^2 \tag{A1}$$

$$= L_{\overline{A}}^2 + L_{A'}^2$$

where





$$L_{\overline{A}} = \left(\frac{1}{N}\sum_{i=1}^{N}\left\|\overline{\boldsymbol{A}}\right\|^2\right)^{1/2} = \left\|\overline{\boldsymbol{A}}\right\| = \left(\sum_{i=1}^{M}\overline{a}_i^2\right)^{1/2}$$

is the RMSL of the mean vector field, and $L_{A'} = \left(\frac{1}{N}\sum_{j=1}^{N}\left\|\boldsymbol{A}'_j\right\|^2\right)^{1/2}$ is the RMSL of the vector anomaly field.

Equation (A1) can be reorganized as:

$$L_A{}^2 = \sum_{i=1}^{M}\left(\frac{1}{N}\sum_{j=1}^{N}\overline{a}_i{}^2\right) + \sum_{i=1}^{M}\left(\frac{1}{N}\sum_{j=1}^{N}{a'_{ij}}^2\right) = \sum_{i=1}^{M}\left(\overline{a}_i{}^2 + {\sigma_{ai'}}^2\right) \tag{A2}$$

where

5  $\sigma_{ai'} = \left(\frac{1}{N}\sum_{j=1}^{N}{a'_{ij}}^2\right)^{1/2}$;     $i = 1, 2, \ldots, M$

is the centered RMS value (or standard deviation) of the $i$-th component of vector field $\boldsymbol{A}$. $L_A$ is the RMSL of vector field $\boldsymbol{A}$ which measures the overall mean value and variance of all components of the vector field.

Similarly, we have

$$L_B{}^2 = L_{\overline{B}}^2 + L_{B'}^2 = \sum_{i=1}^{M}\left(\overline{b}_i{}^2 + {\sigma_{bi'}}^2\right) \tag{A3}$$

where

10  $\sigma_{bi'} = \left(\frac{1}{N}\sum_{j=1}^{N}{b'_{ij}}^2\right)^{1/2}$;     $i = 1, 2, \ldots, M$

is the centered RMS value (or standard deviation) of the $i$-th component of vector field $\boldsymbol{B}$.

With the support of Eq. (13), the VSC can be written as:

$$R_v = \frac{1}{NL_AL_B}\sum_{j=1}^{N}\sum_{i=1}^{M}\left(\overline{a}_i + a'_{ij}\right)\left(\overline{b}_i + b'_{ij}\right)$$

$$= \frac{1}{NL_AL_B}\sum_{j=1}^{N}\sum_{i=1}^{M}\left(\overline{a}_i\overline{b}_i + \overline{a}_ib'_{ij} + \overline{b}_ia'_{ij} + a'_{ij}b'_{ij}\right)$$

$$= \frac{1}{NL_AL_B}\sum_{i=1}^{M}\left(\sum_{j=1}^{N}\overline{a}_i\overline{b}_i + \sum_{j=1}^{N}\overline{a}_ib'_{ij} + \sum_{j=1}^{N}\overline{b}_ia'_{ij} + \sum_{j=1}^{N}a'_{ij}b'_{ij}\right)$$

$$= \frac{L_{\overline{A}}L_{\overline{B}}}{L_AL_B}R_{\overline{v}} + \frac{L_{A'}L_{B'}}{L_AL_B}R_{v'}$$

Given that $\sum_{j=1}^{N}a'_{ij} = \sum_{j=1}^{N}b'_{ij} = 0$ for all $i$ ($1 \leq i \leq M$), we obtain

$$R_v = \frac{1}{NL_AL_B}\sum_{i=1}^{M}\left(\sum_{j=1}^{N}\overline{a}_i\overline{b}_i + + \sum_{j=1}^{N}a'_{ij}b'_{ij}\right)$$

$$= \frac{L_{\overline{A}}L_{\overline{B}}}{L_AL_B}R_{\overline{v}} + \frac{L_{A'}L_{B'}}{L_AL_B}R_{v'} \tag{A4}$$



where

$$R_{\bar{v}} = \frac{1}{NL_{\bar{A}}L_{\bar{B}}} \sum_{j=1}^{N} \sum_{i=1}^{M} \bar{a}_i \bar{b}_i = \frac{1}{L_{\bar{A}}L_{\bar{B}}} \sum_{i=1}^{M} \bar{a}_i \bar{b}_i \tag{A5}$$

$$R_{v'} = \frac{1}{NL_{A'}L_{B'}} \sum_{j=1}^{N} \sum_{i=1}^{M} a'_{ij} b'_{ij} \tag{A6}$$

Given the Cauchy-Schwarz inequality, Eq. (A5) can be rewritten as:

$$R_{\bar{v}}^2 = \frac{\left(\sum_{i=1}^{M} \bar{a}_i \bar{b}_i\right)^2}{(L_{\bar{A}}L_{\bar{B}})^2} \leq \frac{\sum_{i=1}^{M} \bar{a}_i^2 \sum_{i=1}^{M} \bar{b}_i^2}{(L_{\bar{A}}L_{\bar{B}})^2} = \frac{L_{\bar{A}}^2 L_{\bar{B}}^2}{(L_{\bar{A}}L_{\bar{B}})^2} = 1$$

$R_{\bar{v}}$ reaches its maximum value of 1 when $\frac{\bar{a}_1}{\bar{b}_1} = \frac{\bar{a}_2}{\bar{b}_2} = \cdots = \frac{\bar{a}_m}{\bar{b}_m} > 0$. In contrast, $R_{\bar{v}}$ reaches its minimum value of -1 when

$\frac{\bar{a}_1}{\bar{b}_1} = \frac{\bar{a}_2}{\bar{b}_2} = \cdots = \frac{\bar{a}_m}{\bar{b}_m} < 0$. $R_{\bar{v}}$ measures the extent of correlation between modeled and observed mean values across all

5  components of two vector fields.

Eq. (A6) can be rewritten as:

$$
\begin{aligned}
R_{v'} &= \frac{1}{NL_{A'}L_{B'}} \sum_{j=1}^{N} \sum_{i=1}^{M} \frac{a'_{ij} b'_{ij}}{\sigma_{ai'}\sigma_{bi'}} \sigma_{ai'}\sigma_{bi'} \\
&= \frac{1}{NL_{A'}L_{B'}} \sum_{i=1}^{M} \sum_{j=1}^{N} \frac{a'_{ij} b'_{ij}}{\sigma_{ai'}\sigma_{bi'}} \sigma_{ai'}\sigma_{bi'} \\
&= \frac{1}{L_{A'}L_{B'}} \sum_{i=1}^{M} \left( \sigma_{ai'}\sigma_{bi'} \frac{1}{N} \sum_{j=1}^{N} \frac{a'_{ij} b'_{ij}}{\sigma_{ai'}\sigma_{bi'}} \right) \\
&= \frac{1}{L_{A'}L_{B'}} \sum_{i=1}^{M} \sigma_{ai'}\sigma_{bi'} r_i
\end{aligned}
\tag{A7}
$$

where $\sigma_{ai'}$ and $\sigma_{bi'}$ are the centered RMS values (or standard deviation) of the $i$-th component of vector field $\boldsymbol{A}$ and $\boldsymbol{B}$, respectively. $r_i$ represents the centered correlation coefficients between the $i$-th paired components of vector fields $\boldsymbol{A}$ and $\boldsymbol{B}$.

10  $R_{v'}$ can be interpreted as a weighted average of the correlation coefficients across all paired components between two vector fields. The weight coefficients are proportional to the product of standard deviations between paired variables. Clearly, the VSC is simultaneously determined based on the correlation of various mean fields and the overall correlation of anomaly fields across all paired components between two vector fields (Eqs. A4, A5, A7).

The RMSVD between two vector fields can also be represented by the mean and anomaly fields:





$$RMSVD^2 = \frac{1}{N}\sum_{j=1}^{N}\left|\boldsymbol{A}_j - \boldsymbol{B}_j\right|^2$$

$$= \frac{1}{N}\sum_{j=1}^{N}\sum_{i=1}^{M}\left(\bar{a}_i + a'_{ij} - \bar{b}_i - b'_{ij}\right)^2$$

$$= \frac{1}{N}\sum_{i=1}^{M}\sum_{j=1}^{N}\left(\bar{a}_i - \bar{b}_i + a'_{ij} - b'_{ij}\right)^2$$

$$= \frac{1}{N}\sum_{i=1}^{M}\left(\sum_{j=1}^{N}\left(\bar{a}_i - \bar{b}_i\right)^2 + \sum_{j=1}^{N}\left(a'_{ij} - b'_{ij}\right)^2 + 2\left(\bar{a}_i - \bar{b}_i\right)\sum_{j=1}^{N}\left(a'_{ij} - b'_{ij}\right)\right)$$

Given that $\sum_{j=1}^{N} a'_{ij} = \sum_{j=1}^{N} b'_{ij} = 0$ for all $i$ ($1 \le i \le$ M), we obtain

$$RMSVD^2 = \frac{1}{N}\sum_{i=1}^{M}\left(\sum_{j=1}^{N}\left(\bar{a}_i - \bar{b}_i\right)^2 + \sum_{j=1}^{N}\left(a'_{ij} - b'_{ij}\right)^2\right)$$

$$= \frac{1}{N}\sum_{j=1}^{N}\sum_{i=1}^{M}\left(\bar{a}_i - \bar{b}_i\right)^2 + \frac{1}{N}\sum_{j=1}^{N}\sum_{i=1}^{M}\left(a'_{ij} - b'_{ij}\right)^2$$

$$= \frac{1}{N}\sum_{j=1}^{N}\left|\overline{\boldsymbol{A}} - \overline{\boldsymbol{B}}\right|^2 + \frac{1}{N}\sum_{j=1}^{N}\left|\boldsymbol{A}'_j - \boldsymbol{B}'_j\right|^2 \qquad (A8)$$

$$= \left|\overline{\boldsymbol{A}} - \overline{\boldsymbol{B}}\right|^2 + \frac{1}{N}\sum_{j=1}^{N}\left|\boldsymbol{A}'_j - \boldsymbol{B}'_j\right|^2$$

The RMSVD between two vector fields is determined based on the differences in mean vector fields plus the RMSVD of the anomaly vector field. RMSVD$^2$ can further be written as:

$$RMSVD^2 = \left|\overline{A} - \overline{B}\right|^2 + \frac{1}{N}\sum_{j=1}^{N}\left|A'_j - B'_j\right|^2$$

$$= \sum_{i=1}^{M}\left(\bar{a}_i - \bar{b}_i\right)^2 + \frac{1}{N}\sum_{j=1}^{N}\sum_{i=1}^{M}\left(a'_{ij} - b'_{ij}\right)^2 \qquad (A9)$$

$$= \sum_{i=1}^{M}\left(\bar{a}_i - \bar{b}_i\right)^2 + \sum_{i=1}^{M}\left(\frac{1}{N}\sum_{j=1}^{N}\left(a'_{ij} - b'_{ij}\right)^2\right)$$

$$= \sum_{i=1}^{M}\left(\bar{a}_i - \bar{b}_i\right)^2 + \sum_{i=1}^{M} RMSD_i'^2$$





where $RMSD'_i$ is the centered RMSD between the $i$-th paired components of vector fields $\boldsymbol{A}$ and $\boldsymbol{B}$. From the viewpoint of MVIE, the RMSVD can be interpreted as the overall mean difference of all fields plus the overall RMSD of all anomaly fields.

5    The statistics can be computed based on the full vector fields or anomaly vector fields depending on the concern of evaluation. The statistical quantities, i.e., RMSL, VSC, and RMSVD, computed based on the full vector fields represent the uncentered pattern statistics, which include the statistics from both the mean and anomaly fields. Alternatively, three statistics can also be computed based on the anomaly fields, yielding centered statistics, which only measure the anomaly fields. The full vector fields should be used if both the mean and anomaly need to be evaluated. In contrast, the anomaly

10   vector fields should be used if anomaly fields are the primary concern.





**Appendix B: Weighted multivariable integrated evaluation with VFE diagram**

In terms of model evaluation, one may care for some variables more than other variables, although all variables are of great concern. In such a circumstance, it would be useful to weight different variables to make the VSC, RMSL, and RMSVD more sensitive to some variables than to others. Without loss of generality, the weighted- and normalized-vector fields $\boldsymbol{A}^w$ and $\boldsymbol{B}^w$ can be written as a pair of vector sequences:

$$\boldsymbol{A}_j^w = w \cdot \boldsymbol{A}_j^\star = (w_1 a_{1j}^\star, w_2 a_{2j}^\star, \ldots, w_M a_{Mj}^\star); \qquad j = 1, 2, \ldots, N \tag{B1}$$

$$\boldsymbol{B}_j^w = w \cdot \boldsymbol{B}_j^\star = \left(w_1 b_{1j}^\star, w_2 b_{2j}^\star, \ldots, w_M b_{Mj}^\star\right); \qquad j = 1, 2, \ldots, N \tag{B2}$$

where $a_{1j}^\star$ and $b_{1j}^\star$ are the same as in Eqs. (19) and (20). $w_i$ is the weight coefficient for the $i$-th component of the vector field and satisfies the constraint.

$$\sum_{i=1}^M w_i = M$$

Note that the weighting should be applied to the normalized model and observational data (Eqs. B1, B2). Otherwise, the normalization would remove the weight coefficient. Based on Eq. (3), the square of the RMSL of the normalized vector field A⋆ can be written as follows:

$$L_A^{\star\,2} = \sum_{i=1}^M L_{ai}^{\star\,2} \tag{B3}$$

where $L_{ai}^\star = \left(\frac{1}{N}\sum_{j=1}^N a_{ij}^{\star\,2}\right)^{1/2}$ denotes the RMS values of the $i$-th component of the normalized vector field A⋆. Similarly, the RMSL of weighted- and normalized-vector fields can be written as follows:

$$L_A^{w\,2} = \sum_{i=1}^M L_{ai}^{w\,2} \tag{B4}$$

$L_{ai}^w = \left(\frac{1}{N}\sum_{j=1}^N w_i^2 a_{ij}^{\star\,2}\right)^{1/2}$ is the RMS value of the $i$-th component of vector field A$^w$. With the support of Eqs. (B1), (B3), and (B4), it is easy to obtain

$$L_A^{w\,2} = \sum_{i=1}^M L_{ai}^{w\,2} = \sum_{i=1}^M w_i^2 L_{a2}^{\star\,2} \tag{B5}$$

The RMSL of vector field $\boldsymbol{A}^w$ is determined based on the weighted RMS values across all components of the vector field. The contribution of the $i$-th RMS value, $L_{a2}^\star$, to the quadratic RMSL of the vector field is weighted by $w_i^2$. The RMS value accounts for more of the RMSL when its weight coefficient is greater.

Based on Eq. (16), the square of the RMSVD between normalized vector fields **A**⋆ and **B**⋆ can be written as follows:





$$RMSVD^{\star 2} = \sum_{i=1}^{M} RMSD_i^{\star 2} \qquad (B6)$$

where $RMSD_i = \left(\frac{1}{N}\sum_{j=1}^{N}\left(a_{ij}^{\star} - b_{ij}^{\star}\right)^2\right)^{1/2}$ is the RMSD of the $i$-th paired components between normalized vector fields $\boldsymbol{A}^{\star}$ and $\boldsymbol{B}^{\star}$. Similarly, the square of the RMSVD between weighted vector fields $\boldsymbol{A}^{w}$ and $\boldsymbol{B}^{w}$ can be written as follows:

$$RMSVD^{w2} = \sum_{i=1}^{M} (RMSD_i^w)^2 \qquad (B7)$$

$RMSD_i^w = \left(\frac{1}{N}\sum_{j=1}^{N}\left(w_i a_{ij}^{\star} - w_i b_{ij}^{\star}\right)^2\right)^{1/2}$ is the RMSD of the $i$-th paired components between weighted vector fields $\boldsymbol{A}^{w}$ and $\boldsymbol{B}^{w}$. With the aid of Eqs. (B1), (B2), (B6), and (B7), we obtain

$$(RMSVD^w)^2 = \sum_{i=1}^{M} (RMSD_i^w)^2 = \sum_{i=1}^{M} w_i^2 RMSD_i^{\star 2} \qquad (B8)$$

The RMSVD between two vector fields is determined based on the weighted RMSDs across all paired components of two vector fields. The contribution of the $i$-th RMSD to the quadratic RMSVD between two vector fields is weighted by $w_i^2$.

Based on Eq. (13), the VSC between normalized vector fields $\mathbf{A}^{\star}$ and $\mathbf{B}^{\star}$ can be written as follows:

$$R_v^{\star} = \frac{1}{L_A^{\star} \cdot L_B^{\star}} \sum_{i=1}^{M} \sigma_{ai}^{\star} \sigma_{bi}^{\star} R_{ui}^{\star} \qquad (B9)$$

where $\sigma_{ai}^{\star} = \left(\frac{1}{N}\sum_{j=1}^{N} a_{ij}^{\star\,2}\right)^{1/2}$ and $\sigma_{bi}^{\star} = \left(\frac{1}{N}\sum_{j=1}^{N} b_{ij}^{\star\,2}\right)^{1/2}$ are the RMS values for the $i$-th modeled and observational fields.

$R_{ui}^{\star} = \frac{\frac{1}{N}\sum_{j=1}^{N} a_{1j}^{\star} b_{1j}^{\star}}{\sigma_{a1}\sigma_{b1}}$ is the uncentered correlation coefficient for the $i$-th components between two vector fields. Similarly, the VSC between weighted fields can be rewritten as:

$$R_v^w = \frac{1}{L_A^w \cdot L_B^w} \sum_{i=1}^{M} \sigma_{ai}^w \sigma_{bi}^w R_{ui}^w \qquad (B10)$$

where $\sigma_{ai}^w$, $\sigma_{bi}^w$, and $R_{ui}^w$ are the same as $\sigma_{ai}^{\star}$, $\sigma_{bi}^{\star}$, and $R_{ui}^{\star}$, respectively, except they are computed based on the weighted vector fields $\boldsymbol{A}^{w}$ and $\boldsymbol{B}^{w}$. With the aid of Eqs. (B1), (B2), (B9), and (B10), we obtain

$$R_v^w = \frac{1}{L_A^w \cdot L_B^w} \sum_{i=1}^{M} w_i^2 \sigma_{ai} \sigma_{bi} R_{ui} \qquad (B11)$$

The VSC is determined based on the sum of the products of the uncentered correlation coefficients and the RMS values. The

contribution of the $i$-th product term, $\sigma_{ai}\sigma_{bi}R_{ui}$, to the VSC is weighted by $w_i^2$.



**Author contribution**

Z. Xu devised the evaluation method and wrote the paper. All of the authors discussed the results and commented on the manuscript.

5   **Acknowledgements**

We acknowledge the World Climate Research Programme's Working Group on Coupled Modelling, which is responsible for CMIP, and we thank the climate modeling groups for producing and making available their model output. CRU data provided by Climatic Research Unit from their web site at https://crudata.uea.ac.uk/cru/data/hrg/. UDel_AirT_Precip, GPCC Precipitation data, GNCN Gridded V2 data, provided by the NOAA/OAR/ESRL PSD, Boulder, Colorado, USA, from their
10  Web site at http://www.esrl.noaa.gov/psd/. The study was supported jointly by The National Key Research and Development Program of China (2016YFA0600403), the Major Research Plan of the National Science Foundation of China (91637103), and the National Science Foundation of China Grant (41675080, 41675105). This work was also supported by the Jiangsu Collaborative Innovation Center for Climate Change.



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



## Tables

**Table 1.** Multiple statistics of CMIP5 model in simulating surface air temperature and precipitation in terms of climatological mean state and interannual variability. Tm (Pm): climatological mean surface air temperature (precipitation) in summer. Ta (Pa): standard deviation of summer surface air temperature (precipitation). RMS: the ratio of modeled to observed root mean square (RMS) values of the spatial pattern for each variable. corr (RMSD): uncentered spatial correlation coefficient (root mean square deviation) between model and observational fields. RMSL, Rv, RMSVD measure the statistics of two vector fields, which can represent the overall statistics of all fields (Eqs. 11, 14, 21). RMSL was shown as the ratio of model simulated RMSL to the observed RMSL. RMS_stddev is the standard deviation of four RMS values, which describe the dispersion of RMS values of Tm, Pm, Ta, and Pa. MIEI: multivariable integrated evaluation index. Model performance is indicated by the color scale: lighter colors denote better model performance.

| METRICS | RMS | | | | RMSL | CORR | | | | Rv | RMSD | | | | RMSVD | RMS_std | MIEI |
|---|---|---|---|---|---|---|---|---|---|---|---|---|---|---|---|---|---|
| | Tm | Pm | Ta | Pa | | Tm | Pm | Ta | Pa | | Tm | Pm | Ta | Pa | | | |
| Model 1 | 1.01 | 1.00 | 1.15 | 0.98 | 1.04 | 1.00 | 0.92 | 0.96 | 0.92 | 0.95 | 0.10 | 0.41 | 0.35 | 0.39 | 0.33 | 0.08 | 0.34 |
| Model 2 | 1.04 | 1.00 | 1.16 | 1.03 | 1.06 | 0.99 | 0.84 | 0.94 | 0.85 | 0.91 | 0.11 | 0.56 | 0.40 | 0.56 | 0.45 | 0.07 | 0.44 |
| Model 3 | 1.06 | 0.81 | 1.27 | 0.79 | 1.00 | 0.99 | 0.89 | 0.95 | 0.86 | 0.91 | 0.12 | 0.47 | 0.45 | 0.52 | 0.42 | 0.23 | 0.48 |
| Model 4 | 0.97 | 0.86 | 1.21 | 0.74 | 0.96 | 1.00 | 0.91 | 0.96 | 0.87 | 0.92 | 0.09 | 0.42 | 0.38 | 0.50 | 0.38 | 0.20 | 0.44 |
| Model 5 | 0.99 | 0.95 | 1.42 | 1.17 | 1.15 | 0.99 | 0.87 | 0.95 | 0.84 | 0.90 | 0.10 | 0.50 | 0.57 | 0.63 | 0.49 | 0.21 | 0.51 |
| Model 6 | 1.06 | 0.98 | 0.99 | 0.84 | 0.97 | 1.00 | 0.91 | 0.96 | 0.90 | 0.94 | 0.10 | 0.42 | 0.27 | 0.43 | 0.33 | 0.09 | 0.35 |
| Model 7 | 1.08 | 1.10 | 1.11 | 1.07 | 1.09 | 0.99 | 0.90 | 0.95 | 0.90 | 0.94 | 0.13 | 0.47 | 0.36 | 0.47 | 0.39 | 0.02 | 0.37 |
| Model 8 | 1.01 | 0.99 | 1.11 | 1.00 | 1.03 | 1.00 | 0.90 | 0.96 | 0.88 | 0.93 | 0.08 | 0.45 | 0.31 | 0.49 | 0.37 | 0.06 | 0.37 |
| Model 9 | 0.92 | 0.93 | 1.00 | 0.69 | 0.89 | 0.99 | 0.87 | 0.94 | 0.87 | 0.91 | 0.14 | 0.50 | 0.35 | 0.53 | 0.41 | 0.14 | 0.46 |





**Captioned Figures**

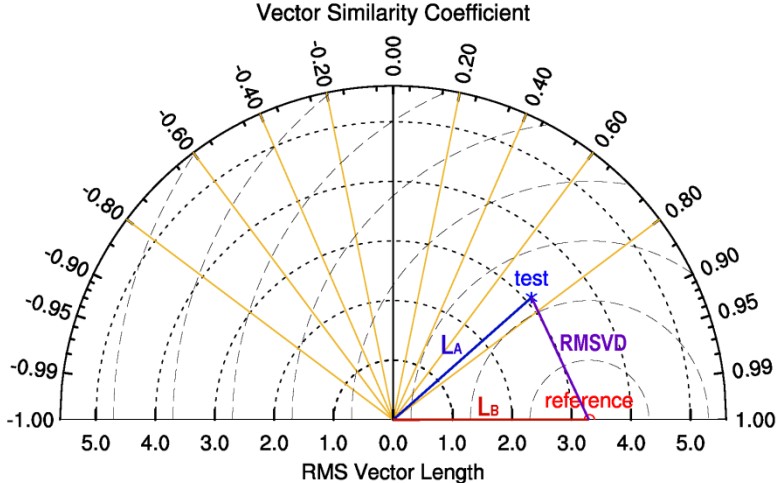

**Figure 1: VFE diagram for displaying multiple statistics of two vector fields. The vector similarity coefficient between two vector fields is given by the azimuthal position of the test field. The radial distance from the origin is proportional to the RMS length. $L_A$ and $L_B$ represent the RMS lengths of the test and reference vector fields, respectively. The RMSVD between the test and reference fields is proportional to their distance apart (dashed contours given in the same units as those for the RMS length).**





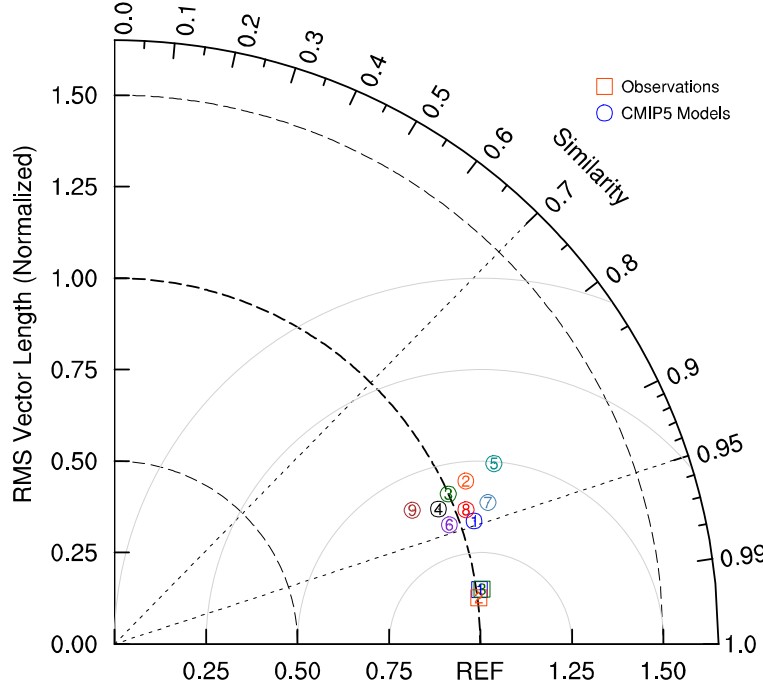

**Figure 2: VFE diagram describing the normalized climatological mean SAT, precipitation, and interannual variabilities of SAT and precipitation over a land area between 60°S–60°N simulated by 9 CMIP5 models compared with three pairs of SAT and precipitation data observed during the period from 1961 to 2000. The RMS length and the RMSVD have been normalized by dividing the RMS length derived from the observed data.**





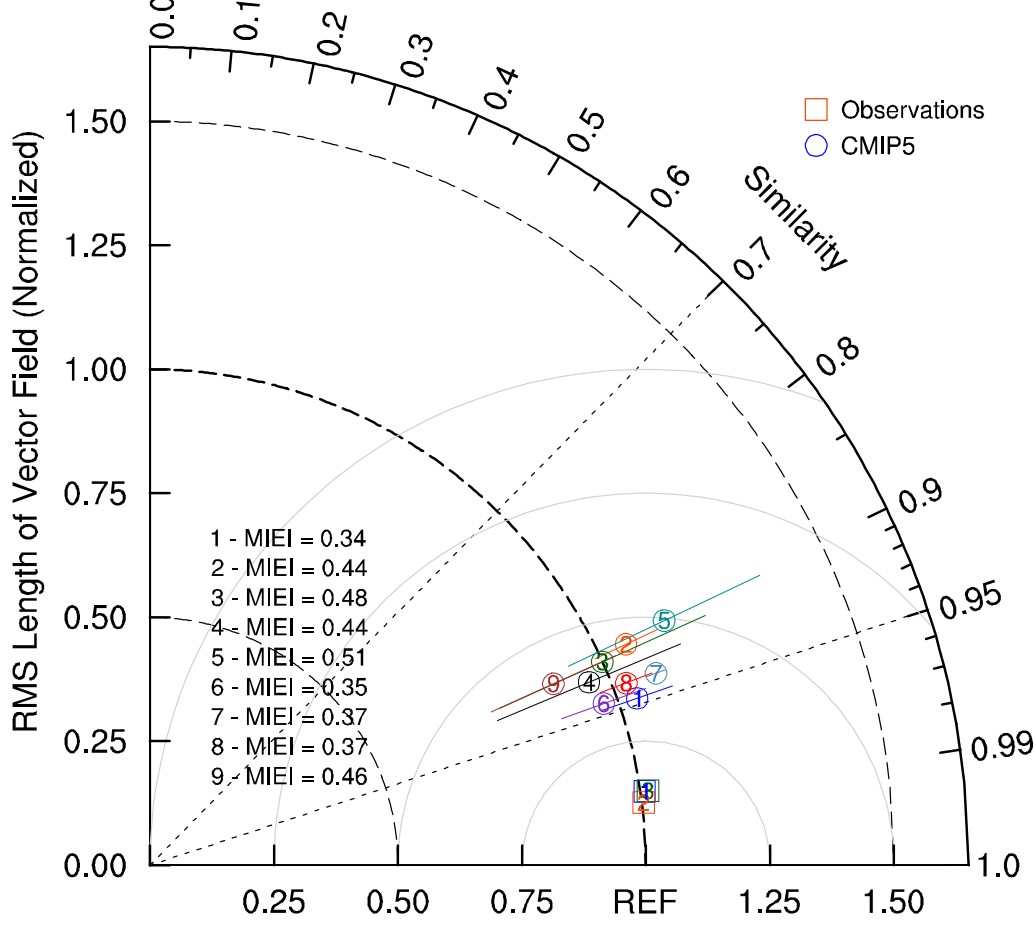

**Figure 3: Same as in Figure 2, except the standard deviations of the RMS values of various fields are shown as line segments. The value of the MIEI for each model is also shown in the diagram.**





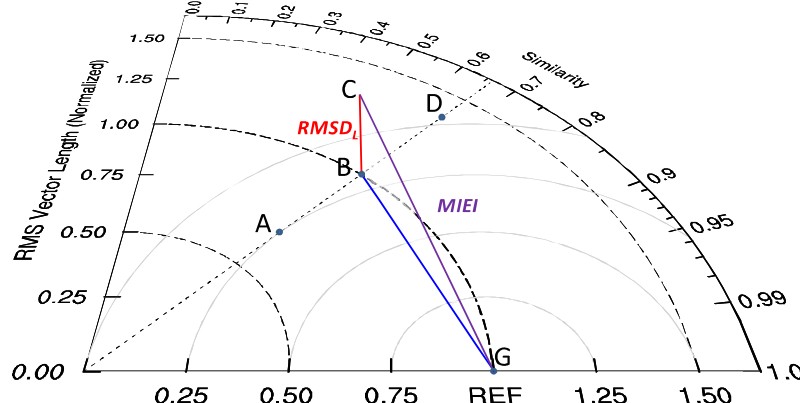

**Figure 4: Schematic diagram displaying the relationship between the RMSVD, RMSD$_L$, and MIEI. The points A, B, and D represent different models. The RMSD$_L$ measures the overall difference between the modeled RMS values and the observed ones. The line segment BC is vertical with respect to the VFE diagram. The length of line segment BG is determined based on the vector field similarity, which measures the overall pattern similarity of various scalar fields relative to the observed ones. Thus, the MIEI index takes both the pattern similarity and the RMS values of various scalar fields into account.**





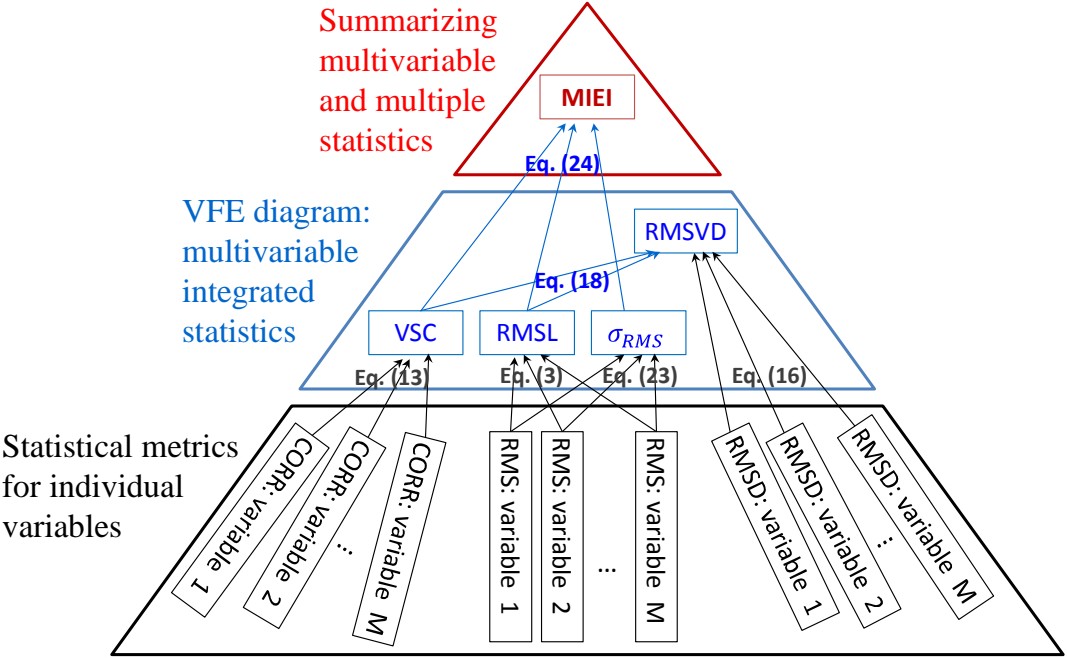

**Figure 5: Pyramid chart showing the relationship between three levels of metrics. The first level of metrics, i.e., correlation coefficient (R), RMS value, and RMSD, measures the model performance in terms of individual variables. The second level of metrics, i.e., VSC, RMSL, variance of RMS values ($\sigma_{RMS}^2$), and RMSVD, is derived from the first level of metrics and summarizes the overall performance of a climate model in simulating multiple fields. The MIEI further summarizes the VSC, RMSL, and $\sigma_{RMS}^2$ into a single index to rank various climate models in terms of simulating multiple fields.**