# Peer review of "Multivariable Integrated Evaluation of Model Performance with the Vector Field Evaluation Diagram"

_Geoscientific Model Development, 2017_

## Referee Comment (RC1) · Anonymous Referee #1 · 14 Jun 2017

**Multivariable Integrated Evaluation of Model Performance with the Vector Field Evaluation Diagram by Z. Xu, Y. Han, and C. Fu**

The authors describe a method to assess the performance of climate models in simulating an arbitrary number of equally important variables based on the concept of the vector field evaluation (VFE) introduced by Xu et al. (2016). In addition, the authors describe a method to collapse the three different metrics root mean square length (RMSL), vector field similarity coefficient, and root mean square vector deviation (RMSVD) into a single index that can be used to rank the models by overall performance for the set of given variables.

The manuscript is generally well written and I suggest minor revisions to the manuscript before publication in Geoscientific Model Development addressing the points given below.

**General comments**

- As an example application of the model evaluation method presented in this paper, three different temperature / precipitation datasets are averaged as reference dataset. Datasets such as the CRU 2m temperature data typically contain missing values (also over land). How are missing values being treated in this study? Also, are the grid cells weighted by their surface area? This would be needed to make sure that the skill scores are representative for the global mean values of the respective quantities. If not, the calculated average metrics would be very hard to interpret as e.g. grid cells in polar region would receive more relative weight than grid cells in low latitudes. As the method presented here suggests evaluation of global averages, I would like to see one or two sentences on this issue. Also, please be more specific on the processing of the observational data regarding e.g. missing values.

- Some parts of the manuscript are somewhat repetitive and could be shortened. For example on p.8, l. 9-10 it reads "Thus, three statistical quantities can be indicated by a single point on the VFE diagram", which is almost identical to p. 10, l. 10-12: "[...] the three quantities [...] can be represented by a single point for each model on the VFE diagram." or to p. 10, l. 13-15: "Thus, each point on the VFE diagram can represent the overall performance of an individual model in terms of 3 statistical quantities [...]". I suggest to go through the text and remove

such repetitions where possible.

- The authors propose to include the standard deviation of the RMS values of multiple scalar fields into the VFE diagram as an additional performance measure (p. 11, l. 12-14). It remains unclear to me whether the length of the proposed additional line segments in figure 3 are $\sigma_{RMS}$ or actually $\pm\sigma_{RMS}$, i.e. $2\sigma_{RMS}$. Or did the authors mean variance of the RMS values (equation 23)? Please clarify.

- A proper evaluation of the performance of climate models usually requires to take into account observational uncertainties. Differences between models and observations can only be interpreted as model errors or lack of model skill if the differences are larger than the observational uncertainty. This is particularly the case for variables with a large uncertainty such as, for instance, ice water path, but also important when ranking models by performance. More and more observational datasets provide estimates of the observational uncertainty. What are the authors' thoughts about including such additional information into their calculations, in particular when calculating skill scores such as the presented multivariable integrated evaluation index (MIEI) that is then used to rank models according to their average performance skill?

**Specific comments**

- p. 1, l. 21: "[...] evaluation *of* model performance."

- p. 10, l. 5: "what is meant by "summer SAT and precipitation"? Is this an average over the months June, July, August? Please be more specific.

- p. 12, l. 20, "*In comparison with* the RMSVD, [...]": did you mean "In contrast to [...]"?

- p. 13, l. 2, "Index" → "index"

- p. 13, l. 5, "[...] but a larger MIEI *relative to* [...]": did you mean "compared to"?

- p. 26, l. 3, "CMIP5 model" → "CMIP5 model*s*"

- p. 26, table 1: it would be interesting to add the performance of the individual observational datasets to the table as a "rough estimate of the observational uncertainties" as stated on p. 9, l. 22-23.

- p. 27, caption of figure 1, l. 4: delete "apart"

- figure 2 is fully included in figure 3 and could be deleted

- p. 31, figure 5: the second level of metrics includes $\sigma_{RMS}$ while the caption and the referenced equation 23 specify the variance of RMS values ($\sigma^2_{RMS}$). Which one is correct? Is there a "$^2$" missing?

---

## Referee Comment (RC2) · Anonymous Referee #2 · 1 Jul 2017

This paper is closely related to an earlier paper published by the authors in GMD (doi:10.5194/gmd-9-4365-2016), but in this new paper I believe there is a fatal flaw in that the "vectors" considered are constructed from components representing individual fields which are in general not independent. The fields produced by climate models (and indeed the fields observed in the physical world) are rarely truly independent. Consider, for example, the trivial case of temperature field at 900 hPa and the temperature field at 850 hPa. These fields would be very similar (with second one being slightly cooler than the first), and since they are not independent, they are unsuitable for use as components of a vector. Similarly there are relationships between specific humidity and temperature that yield high correlations between them. Thus, the "vectors" defined in this paper are based on dimensions that are not independent (i.e., not

orthogonal).

If I am correct that orthogonality of the vector components is a requirement, then the paper rests on unsound mathematics and should be rejected.

If I am wrong, then the paper should be considered, but I'm not sure it adds much to what was already published in the earlier paper where the vector components were based on spatial direction (rather than variable). Isn't the present paper an obvious extension of the earlier paper (simply an application to a different vector, one based on dimensions defined by variables rather than spatial dimensions)?

---

## Author Comment (AC2) · 28 Jul 2017

We would like to thank the reviewer's comments. Our point-by-point responses are listed below:

==========================

Reviewer #2

This paper is closely related to an earlier paper published by the authors in GMD (doi:10.5194/gmd-9-4365-2016), but in this new paper I believe there is a fatal flaw in that the "vectors" considered are constructed from components representing individual fields which are in general not independent. The fields produced by climate models (and indeed the fields observed in the physical world) are rarely truly indepen-

dent. Consider, for example, the trivial case of temperature field at 900 hPa and the temperature field at 850 hPa. These fields would be very similar (with second one being slightly cooler than the first), and since they are not independent, they are un-suitable for use as components of a vector. Similarly there are relationships between specific humidity and temperature that yield high correlations between them. Thus, the "vectors" defined in this paper are based on dimensions that are not independent (i.e., not orthogonal). If I am correct that orthogonality of the vector components is a requirement, then the paper rests on unsound mathematics and should be rejected.

RESPONSE:

The definitions of the statistical quantities, i.e., RMSL, Rv, RMSVD, do request an orthogonal coordinates. In our study, we use the M-dimensional standard Cartesian coordinate. The Cartesian axes are mutually perpendicular to each other. Therefore, the axes are always orthogonal no matter whether the scalar fields are independent or not. We just assign each scalar variable to the corresponding component of the orthogonal coordinates and compare two constructed vector fields.

The statistics are still meaningful when the components of vector field are dependent. Consider an idealized case, assuming we constructed a modeled 2-dimensional vector fields (A) with exactly same x- and y- components (the x- and y- components are fully dependent). Consequently, all vectors in the vector fields point in the same direction (In contrast, the vectors point in different directions if the x- and y- components are not fully dependent). Assuming we also constructed an observed 2-dimensional vector fields (B) with exactly same x- and y- component, under such a circumstance, the vector similarity coefficient, Rv, between A and B is equal to the Pearson's correlation coefficient between the x- (or y-) component of A and B. This is still a meaningful measurement on the similarity of two vector fields even if the x- and y-components are fully dependent. As all vectors point in the same direction, one can only consider the variation of vector length. Under such a circumstance, the vector field can be treated as a scalar field (vector length) and one can use correlation coefficient to measure the

relationship of two scalar fields, which is consistent with the vector similarity coefficient. Similarly, RMSL and RMSVD do not request the independence of various scalar fields, either.
* * *
If I am wrong, then the paper should be considered, but I'm not sure it adds much to what was already published in the earlier paper where the vector components were based on spatial direction (rather than variable). Isn't the present paper an obvious extension of the earlier paper (simply an application to a different vector, one based on dimensions defined by variables rather than spatial dimensions)?

RESPONSE:

Two papers do rest on the same mathematics except that the second paper generalizes the VFE diagram to evaluate vector fields in arbitrary dimensions against the vector fields in two-dimensions in the earlier paper. However, two papers present very different aspects of model evaluation. The earlier paper presented an evaluation method of model performance in simulating individual vector field, e.g., vector winds, temperature gradient. The current paper reports a multi-variable integrated evaluation (MVIE) method of model performance.

The scientific merits of this study relative to previous model evaluation methods mainly includes the following aspects:

(1) Our study provides a more comprehensive way to evaluate model performance in terms of multiple fields compared to the previous methods. For example, as discussed in the introduction section, the commonly used multivariable evaluation method, i.e. the portrait diagram (MCPI, Gleckler et al., 2008), only considers the root mean square errors (RMSEs) of various fields. Although RMSE takes both the correlation coefficient and standard deviation into account, the RMSE cannot explicitly measure the pattern similarity and variance. In contrast, the MVIE method includes multiple statistical quantities, e.g., RMSL, VSC, and RMSVD, can explicitly measure the pattern similarity, mean and variance, and the overall difference between two constructed vector fields. Therefore, the MVIE method can provide a more comprehensive evaluation of model performance.

(2) As shown in the pyramid chart (Fig. 4), the second paper constructs a hierarchical evaluation framework of model performance which is composed of commonly used statistical metrics for individual variables, VFE diagram, and multivariable integrated evaluation index (MIEI). The first level of metrics, i.e., correlation coefficient (R), RMS value, and RMSD, measures the model performance in terms of individual variables. The first level of metrics can provide more detailed information in specific aspect of model performance but is lack of summarization and cannot provide a quantitative evaluation on model performance in simulating multiple fields. The second level metrics, i.e., VSC, RMSL, standard deviation of RMS values ($\sigma$RMS), and RMSVD in the VFE diagram, is derived from the first level of metrics and summarizes the overall performance of a climate model in simulating multiple fields. The VFE diagram well summarized multiple statistics but is still hard to rank model performance. The MIEI further summarizes the VSC, RMSL, and $\sigma$RMS in the VFE diagram into a single index to rank various climate models in terms of the performance in simulating multiple fields. Thus, the hierarchical evaluation framework allows one to choose different levels of metrics, which will facilitate the evaluation and inter-comparison of model performance.

The abovementioned two aspects are new and constitute significant advances relative to previous model evaluation methods. Our first paper published in GMD in 2016 did not tackle any of abovementioned aspects. Instead, the first paper focused on the evaluation of model performance in simulating individual vector field, e.g., vector winds.

---

## Author Response (AR1)

We would like to thank two reviewers for their valuable comments. Our point-by-point responses to the comments are detailed in the following pages. Reviewers' comments are in black font. Our responses are in orange font.

5  Reviewer #1

Multivariable Integrated Evaluation of Model Performance with the Vector Field Evaluation Diagram by Z. Xu, Y. Han, and C. Fu

10 The authors describe a method to assess the performance of climate models in simulating an arbitrary number of equally important variables based on the concept of the vector field evaluation (VFE) introduced by Xu et al. (2016). In addition, the authors describe a method to collapse the three different metrics root mean square length (RMSL), vector field similarity coefficient, and root mean square vector deviation (RMSVD) into a single index that can be
15 used to rank the models by overall performance for the set of given variables. The manuscript is generally well written and I suggest minor revisions to the manuscript before publication in Geoscientific Model Development addressing the points given below.

General comments
20 As an example application of the model evaluation method presented in this paper, three different temperature / precipitation datasets are averaged as reference dataset. Datasets such as the CRU 2m temperature data typically contain missing values (also over land). How are missing values being treated in this study? Also, are the grid cells weighted by their surface area? This would be needed to make sure that the skill scores are representative for
25 the global mean values of the respective quantities. If not, the calculated average metrics would be very hard to interpret as e.g. grid cells in polar region would receive more relative weight than grid cells in low latitudes. As the method presented here suggests evaluation of global averages, I would like to see one or two sentences on this issue. Also, please be more specific on the processing of the observational data regarding e.g. missing values.
30 RESPONSE:
The CRU dataset used is CRUTSv3.24 which was constructed to "provide full coverage of the specified continental land area, with no missing data (Harris et al., 2014)". We checked the data and confirmed that there is no missing value over the land area. The data was assigned to climatological mean value if missing data presents during the construction of CRU datasets

(Harris et al., 2014). We also checked the other two pairs of temperature and precipitation datasets; there are no missing data on the continents, either.

Harris et al., 2014: Updated high-resolution grids of monthly climatic observations – the CRU TS3.10 Dataset. Int. J. Climatol. 34: 623–642.

The grid cells were not weighted by their surface area in the previous manuscript. We agree with the reviewer that area weighting should be considered here. In the revised manuscript (P9, L22-23), we take area weighting into account to make the statistics be representative for the global mean values as the reviewer suggested. All related figures and tables are updated. Thanks for the comment!

Some parts of the manuscript are somewhat repetitive and could be shortened. For example on p.8, l. 9-10 it reads "Thus, three statistical quantities can be indicated by a single point on the VFE diagram", which is almost identical to p.10, L. 10-12: "[...] the three quantities [...] can be represented by a single point for each model on the VFE diagram." or to p.10, L. 13-15: "Thus, each point on the VFE diagram can represent the overall performance of an individual model in terms of 3 statistical quantities [...]". I suggest to go through the text and remove such repetitions where possible.

RESPONSE:

The first sentence is retained in the revised manuscript. The second sentence is reworded as "*Thus, each model's performance in simulating multiple variables can be summarized by a single point that is determined by 12 statistical quantities (4 variables × 3 statistics) those derived from various scalar fields*" (P10, L6-8). The third sentence is deleted. We go through the whole manuscript and remove or rewrite a few other sentences those read repetitive. In addition, we replace $\sigma_{ai}$ in Eq. 13 with $L_{ai}$ because the definition of $\sigma_{ai}$ is the same as $L_{ai}$ (Eq. 5).

The authors propose to include the standard deviation of the RMS values of multiple scalar fields into the VFE diagram as an additional performance measure (p. 11, l. 12-14). It remains unclear to me whether the length of the proposed additional line segments in figure 3 are $\sigma_{RMS}$ or actually $\pm\sigma_{RMS}$, i.e. $2\sigma_{RMS}$. Or did the authors mean variance of the RMS values (equation 23)? Please clarify.

RESPONSE:

In the revised manuscript, we clarified "*The length of the line segment is equal to twice the standard deviation of RMS values of multiple scalar fields*" (P11, L5).
For consistency purpose, equation 23 is rewritten in the form of standard deviation rather than variance, because standard deviation is used in Figs. 2, 4.

A proper evaluation of the performance of climate models usually requires to take into account observational uncertainties. Differences between models and observations can only be interpreted as model errors or lack of model skill if the differences are larger than the observational uncertainty. This is particularly the case for variables with a large uncertainty

10  such as, for instance, ice water path, but also important when ranking models by performance. More and more observational datasets provide estimates of the observational uncertainty. What are the authors' thoughts about including such additional information into their calculations, in particular when calculating skill scores such as the presented multivariable integrated evaluation index (MIEI) that is then used to rank models according to their average

15  performance skill?

RESPONSE:
Thanks for the valuable comment. We further discuss how to evaluate the impact of observational uncertainties on model evaluation and ranking in the revised manuscript. To

20  take the advantage of observational uncertainty, one can generate a number of ensemble members of observational estimates using the estimates of the observational uncertainties. Some datasets, e.g., HadCRUT4, already provided such ensemble observational estimates.

We add a new paragraph to discuss how to take observational uncertainty into account in our

25  model evaluation methods. The new paragraph added to the revised manuscript is pasted below:
*"The issue of how to take the observational uncertainties into account is of particular importance in model evaluation and ranking, especially when more and more observational datasets provide estimates of the observational uncertainty. The statistics derived from each*

30  *group of observational estimates are also shown in Table 1, which can roughly quantify the observational uncertainties and its impact on model evaluation. Generally, the colours are clearly lighter for the statistics of individual observed variables in contrast to the modelled variables (Table 1). This indicates that the observational uncertainties are relatively small and should have less impact on the evaluation of model performance. To further quantify the*

*impacts of observational uncertainty on ranking model performance, we calculate the MIEIs of various climate models by taking each group of observational estimates as the reference data. Three groups of observational estimates generate three groups of MIEIs. Afterwards, we calculate Spearman's rank correlation coefficient of each group of MIEIs with those derived from models and ensemble mean of multiple observational estimates. The Spearman's rank correlation coefficients are 0.996, 0.996, and 0.904, respectively, suggesting that the ranks are very close to each other no matter which group of observational estimates is used as reference data. Thus, the observational uncertainty should have less impact on ranking model performance in this case. One can use the average of Spearman's rank correlation coefficients to quantify the consistency of various ranks when a number of observational estimates are available."*

Specific comments

p. 1, l. 21: "[...] evaluation of model performance."

RESPONSE: Done

p. 10, l. 5: "what is meant by "summer SAT and precipitation"? Is this an average over the months June, July, August? Please be more specific.

RESPONSE: Yes, the summer SAT and precipitation refer to the average over the months June, July, and August. This has been clarified in P9, L26 and the caption of Table 1.

p. 12, l. 20, "In comparison with the RMSVD, [...]": did you mean "In contrast to [...]"?

RESPONSE: Yes, and "In comparison with" was replaced with "In contrast to"

p. 13, l. 2, "Index"→"index"

RESPONSE: Done

p. 13, l. 5, "[...] but a larger MIEI relative to [...]": did you mean "compared to"?

RESPONSE: Yes, and "relative to" was replaced with "compared to"

p. 26, l. 3, "CMIP5 model" →"CMIP5 models"

RESPONSE: Done

p. 26, table 1: it would be interesting to add the performance of the individual observational datasets to the table as a "rough estimate of the observational uncertainties" as stated on p. 9, l. 22-23.

RESPONSE: The performance of the individual observational datasets is added to Table 1 in the revised manuscript. P13, L8-12.

p. 27, caption of figure 1, l. 4: delete "apart"

RESPONSE: Done

Figure 2 is fully included in figure 3 and could be deleted

RESPONSE: Figure 2 is deleted in the revised manuscript.

p. 31, figure 5: the second level of metrics includes $\sigma_{RMS}$ while the caption and the referenced equation 23 specify the variance of RMS values ($\sigma^2_{RMS}$). Which one is correct? Is there a "2" missing?

RESPONSE: We rewrite Eq. 23 in the revised manuscript. $\sigma_{RMS}$, instead of $\sigma^2_{RMS}$, is used because the line segments on Fig.3 represent twice the standard deviation of RMS values. The text and figure captions are also updated accordingly.

==========================================
Reviewer #2
This paper is closely related to an earlier paper published by the authors in GMD (doi:10.5194/gmd-9-4365-2016), but in this new paper I believe there is a fatal flaw in that the "vectors" considered are constructed from components representing individual fields which are in general not independent. The fields produced by climate models (and indeed the fields observed in the physical world) are rarely truly independent. Consider, for example, the trivial case of temperature field at 900 hPa and the temperature field at 850 hPa. These fields would be very similar (with second one being slightly cooler than the first), and since they are not independent, they are unsuitable for use as components of a vector. Similarly there are relationships between specific humidity and temperature that yield high correlations between them. Thus, the "vectors" defined in this paper are based on dimensions that are not independent (i.e., not orthogonal). If I am correct that orthogonality of the vector components is a requirement, then the paper rests on unsound mathematics and should be rejected.

The definitions of the statistical quantities, i.e., RMSL, Rv, RMSVD, do request an orthogonal coordinate. In our study, we use the M-dimensional standard Cartesian coordinate. The coordinate axes are mutually perpendicular to each other. Therefore, the coordinate axis is always orthogonal no matter whether or not the scalar fields are independent. We just assign each scalar variable to the corresponding component of the orthogonal coordinate and compare two constructed vector fields.

The statistics are still meaningful when the components of vector field are dependent. Consider an idealized case, assuming we constructed a modeled 2-dimensional vector fields (*A*) with exactly same x- and y- components (the x- and y- components are fully dependent). Consequently, all vectors in the vector fields point in the same direction (In contrast, the vectors point in different directions if the x- and y- components are not fully dependent). Assuming we also constructed an observed 2-dimensional vector fields (*B*) with exactly same x- and y- component, under such a circumstance, the centered vector similarity coefficient, Rv, between *A* and *B* is equal to the Pearson's correlation coefficient between the x- (or y-) component of *A* and *B*. This is still a meaningful measurement on the similarity of two vector fields even if the x- and y-components are fully dependent. As all vectors point in the same direction, one can only consider the variation of vector length. Under such a circumstance, the vector field can be treated as a scalar field (vector length) and one can use correlation coefficient to measure the relationship of two scalar fields, which is consistent with the vector similarity coefficient. Similarly, RMSL and RMSVD do not request the independence of various scalar fields, either.

If I am wrong, then the paper should be considered, but I'm not sure it adds much to what was already published in the earlier paper where the vector components were based on spatial direction (rather than variable). Isn't the present paper an obvious extension of the earlier paper (simply an application to a different vector, one based on dimensions defined by variables rather than spatial dimensions)?

Two papers do rest on the same mathematics except that the second paper generalizes the VFE diagram to evaluate vector fields in arbitrary dimensions against the vector fields in two-dimensions in the earlier paper. However, two papers present very different aspects of model evaluation. The earlier paper presented an evaluation method of model performance in terms of individual vector field, e.g., vector winds, temperature gradient. The current paper reports a multi-variable integrated evaluation (MVIE) method of model performance.

The scientific merit of this study compared with previous model evaluation methods mainly includes the following aspects:

(1) Our study provides a more comprehensive way to evaluate model performance in terms of multiple fields compared to the previous methods. For example, as discussed in the introduction section, the commonly used multivariable evaluation method, i.e. the portrait diagram (MCPI, Gleckler et al., 2008), only considers the root mean square errors (RMSEs) of various fields. Although RMSE takes both the correlation coefficient and standard deviation into account, the RMSE cannot explicitly measure the pattern similarity and variance. In contrast, the MVIE method includes multiple statistical quantities, e.g., RMSL, VSC, and RMSVD, can explicitly measure the pattern similarity, mean and variance, and the overall difference between two constructed vector fields. Therefore, the MVIE method can provide a more comprehensive evaluation of model performance.

(2) As shown in the pyramid chart (Fig. 5), the second paper constructs a hierarchical evaluation framework of model performance which is composed of commonly used statistical metrics for individual variables, VFE diagram, and multivariable integrated evaluation index (MIEI). The first level of metrics, i.e., correlation coefficient (R), RMS value, and RMSD, measures the model performance in terms of individual variables. The first level of metrics can provide detailed information in specific aspect of model performance but is lack of summarization and cannot provide a quantitative evaluation on model performance in simulating multiple fields. The second level metrics, i.e., VSC, RMSL, standard deviation of RMS values ($\sigma_{RMS}$), and RMSVD in the VFE diagram, are derived from the first level of metrics and summarizes the overall performance of a climate model in simulating multiple fields. The VFE diagram well summarized multiple statistics but is still hard to rank model performance. The MIEI further summarizes the VSC, RMSL, and $\sigma_{RMS}$ in the VFE diagram into a single index to rank various climate models in terms of the performance in simulating multiple fields. Our results suggest that the MIEI satisfy the monotonic property of the index with respect to model performance. Therefore it is better suited for ranking model performance than the RMSVD or commonly used model climate performance index (MCPI, (Gleckler et al. 2008). The hierarchical evaluation framework proposed in this study allows one to choose different levels of metrics, which will facilitate the evaluation and inter-comparison of model performance.

The abovementioned two aspects are new and constitute significant advances compared with previous model evaluation methods. Our first paper published in GMD in 2016 did not tackle any of abovementioned aspects. Instead, the first paper focused on the evaluation of model performance in simulating individual vector field, such as vector winds. Therefore, the first and

second papers tackled different aspects of model evaluation although they reset on the same mathematics.

=========================================

In addition to the revisions in response to the reviewers' comments, we also made some minor revisions to the manuscript, e.g., correcting a few typos, rearranging several equations. All revisions can be tracked in the following marked-up manuscript.

*Revised Manuscript with marked-up changes*

[revised manuscript text omitted]

---

## Author Response (AR2)

Topical Editor Comments are copied below:

before I accept your paper finally, I would like you to clarify the first part of your reply to Reviewer #2. As I see it, the question whether the fields to be evaluated are orthogonal or not is irrelevant, both for your old method and for the new one. Non-orthogonality would be fatal if your "vectors" would be used as components of a vector space with linear transformations involving matrix operations. This is not the case, hence orthogonality is not a requirement for your method. In this sense, it seems that the first paragraph of your reply to reviewer #2 is incorrect. You write "The definitions of the statistical quantities ... do request an orthogonal coordinate". I think, it does not!

To my opinion, the use of the word "vector" for something that is merely a "neutral" collection of quantities (a M-tupel), contributes to the puzzling. A "vector" in the physical and mathematical sense is much richer a structure than just a M-tupel.

Please comment to this and include a short paragraph in your manuscript so that the issue is clear to readers without referring to the discussions. The needed paragraph can appear either in the beginning of section 2 or 3.

Resoponse:

Dear Topical Editor,

Thank you so much for your insightful comments. As you mentioned, our method does not require the independence of the fields to be evaluated. In our response to reviewer #2, we were also trying to address the same point. However we may fail to state it clearly. The calculations of RMSL, VSC, and RMSVD involve vector length and inner product of two vectors, which do require an orthogonal coordinate system. In our method, the orthogonal coordinate system is predefined and the axes are always perpendicular to each other. The variables to be evaluated are represented by coordinate values of individual axes. The definition of RMSL, SVC, and RMSVD do not require the independence of the coordinate values between different components of vector fields.

As argued in our previous response to reviewer #2, the statistical quantities, RMSL, SVC, and RMSVD, are still meaningful even some variable are dependent to each other. If some variables to be evaluated are dependent to each other, e.g. skin temperature and surface air temperature, one may want to weight these variables properly because the dependent variables contain redundant information. Consequently, the evaluation may overestimate the importance of dependent variables. The caveat has been included in the summary and discussion section P15, L6-9.

We added a few sentences in P3 L12-14 to make the definition of the vector fields more clear, which should favour the understanding of the definition of RMSL, SVC, and RMSVD and expected to avoid misunderstanding. We also added a few sentences in P8 L14-18 to clarify that our method has no requirement for the independence of the fields to be evaluated.

In addition to the abovementioned changes, we also made a few minor changes to the manuscript. All the changes can be tracked in the following marked-up manuscript.

[revised manuscript text omitted]
} \mathbf{A}_j \cdot \mathbf{B}_j}{\sqrt{\sum_{j=1}^{N} \|\mathbf{A}_j\|^2} \sqrt{\sum_{j=1}^{N} \|\mathbf{B}_j\|^2}} \tag{7}$$

The normalized vectors are written as:

$$\mathbf{A}_j^* = \frac{\mathbf{A}_j}{L_A} = (a_{1j}^*, a_{2j}^*, \dots, a_{Mj}^*) \ ; \qquad j = 1, 2, \dots, N$$

$$\mathbf{B}_j^* = \frac{\mathbf{B}_j}{L_B} = (b_{1j}^*, b_{2j}^*, \dots, b_{Mj}^*) \ ; \qquad j = 1, 2, \dots, N$$

With the aid of Eqs. (1) and (2), we have

$$\sum_{j=1}^{N} \|\mathbf{A}_j^*\|^2 = \sum_{j=1}^{N} \|\mathbf{B}_j^*\|^2 = N \tag{8}$$

We can also represent Eq. (7) in the following form:

$$R_v = \frac{1}{N} \sum_{j=1}^{N} \mathbf{A}_j^* \cdot \mathbf{B}_j^*$$

$$= \frac{1}{N} \sum_{j=1}^{N} \sum_{i=1}^{M} a_{ij}^* b_{ij}^* \tag{9}$$

VSC can be interpreted as the mean of inner products between normalized- and paired-vectors $\mathbf{A}_j^*$ and $\mathbf{B}_j^*$. The squared Euclidean Distance (SED) between $\mathbf{A}_j^*$ and $\mathbf{B}_j^*$ is defined as follows:

[revised manuscript text omitted]